# Information - based uncertainty decomposition in dual channel microwave remote sensing of soil moisture

Bonan Li[1,2], Stephen P. Good[1,2]

[1]Department of Biological & Ecological Engineering, Oregon State University, Corvallis, OR 97330, USA
[2]Water Resources Graduate Program, Oregon State University, Corvallis, OR 97330, USA
*Correspondence to*: Bonan Li (libon@oregonstate.edu)

**Abstract**. NASA's Soil Moisture Active-Passive (SMAP) mission characterizes global spatiotemporal patterns in surface soil moisture using dual L-band microwave retrievals of horizontal ($T_{Bh}$) and vertical ($T_{Bv}$) polarized microwave brightness temperatures through a modeled mechanistic relationship between vegetation opacity, surface scattering albedo, and soil effective temperature ($T_{eff}$). Although this model has been validated against *in situ* soil moisture, there is a lack of systematic characterization of where and why SMAP estimates deviate from the *in situ* observations. Here, we assess how the information content of *in situ* soil moisture observations from the US Climate Reference Network contrasts with (1) the information contained within raw SMAP observations (i.e., 'informational random uncertainty') derived from $T_{Bh}$, $T_{Bv}$ and $T_{eff}$ themselves, and (2) with the information contained in SMAP's Dual Channel Algorithm (DCA) soil moisture estimates (i.e., 'informational model uncertainty') derived from the model's inherent structure and parameterizations. The results show that, on average, 80% of the information in the *in situ* soil moisture is unexplained by SMAP DCA soil moisture estimates. 35% of the unexplained information is caused by the loss of information in the DCA modeling process while the remainder is induced by a lack of additional explanatory power within $T_{Bh}$, $T_{Bv}$ and $T_{eff}$. Overall, retrieval quality of SMAP DCA soil moisture, denoted as the Pearson correlation coefficient between SMAP DCA soil moisture and *in situ* soil moisture, is negatively correlated with the informational uncertainties, with slight differences across different landcovers. The informational model uncertainty (Pearson correlation of -0.59) was found to be more influential than the informational random uncertainty (Pearson correlation of -0.34), suggesting that the poor performance of SMAP DCA at some locations is driven by model parameterization and/or structure and not underlying satellite measurements of $T_{Bh}$ and $T_{Bv}$. A decomposition of mutual information between $T_{Bh}$, $T_{Bv}$ and DCA soil moisture shows that on average 58% of information provided by $T_{Bh}$ and $T_{Bv}$ to DCA estimates is redundancy. The amount of information redundantly and synergistically provided by $T_{Bh}$ and $T_{Bv}$ was found to be tightly related (Pearson correlation of 0.79 and -0.82, respectively) to the retrieval quality of SMAP DCA. The DCA retrieval quality improves as $T_{Bh}$ and $T_{Bv}$ provide more redundant information for the DCA soil moisture. This suggests that the informational redundancy and synergy between these remotely sensed observations can be indicative about soil moisture retrieval quality. This study provides a baseline approach that can also be applied to evaluate other remote sensing models and understand informational loss as satellite retrievals are translated to end user products.

# 1 Introduction

Accurate information on soil moisture is of great importance for understanding various biophysical processes in hydrology, agronomy, and ecosystem sciences (Bassiouni et al., 2020; Uber et al., 2018). The poor spatial representativeness of *in-situ* soil moisture sensors, combined with their labor-intensive installation and maintenance, impedes the application these sensors to understand large scale ecosystem phenomena (Babaeian et al., 2019; Petropoulos et al., 2015). Spaceborne passive microwave remote sensing has been developed as a reliable method to estimate surface soil moisture at large scales (Wigneron et al., 2017). It leverages the large discrepancies in dielectric properties between liquid water and dry soil that result in a high dependency of soil dielectric constants on soil moisture (Njoku and Entekhabi, 1996). Various microwave frequencies have been available to date, amongst which the L-band microwave frequencies were found to be desirable for soil moisture estimations because they can sense soil moisture at a relatively deeper layer (~5cm) and can provide greater vegetation penetration power (Mohanty et al., 2017). Though microwave remote sensing has been investigated for decades, significant uncertainties still exist in both microwave radiometry and in the algorithms used to translate microwave observations to soil moisture estimates (Gruber et al., 2020).

Passive L-band remote sensing soil moisture estimation uses a radiometer to measure surface emission intensity, which is proportional to the brightness temperature (Wang and Qu, 2009). The brightness temperature is linked with soil moisture and vegetation opacity through the '*tau-omega*' emission model and parameterized by soil and vegetation functions (Jackson et al., 1982; Mo et al., 1982). The '*tau-omega*' model rationale has been adopted by NASA's Soil Moisture Active-Passive (SMAP) mission, which is one of the earth observation missions dedicated to estimate soil moisture at L-band microwave frequency (Entekhabi et al., 2010). SMAP implemented two primary algorithms: (1) the single channel algorithm (SCA) that uses one polarized brightness temperature as primary input to retrieve soil moisture and (2) the dual channel algorithm (DCA) that retrieves soil moisture and vegetation opacity simultaneously by taking the polarized brightness temperature information in both horizontal and vertical directions (O'Neill et.al., 2020a). There is strong interest in the DCA approach because of its independent estimation of vegetation opacity in lieu of the specified vegetation climatology employed by the SCA (O'Neill et.al., 2020a). Other L-band focused satellite mission such as Soil Moisture and Ocean Salinity (SMOS) retrieves both soil moisture and vegetation optical depth by using numerous brightness measurements for different incidence angles (Kerr et al., 2012). Additionally, it has been suggested that using a time-integrated vegetation opacity, as is employed in the multi-temporal dual channel algorithm (MT-DCA) for instance (Konings et al., 2016), improves the estimates of soil and vegetation state. These contrasting approaches, as well as other studies on SMAP's temporal polarized ratio algorithm (TPRA) (Gao et al., 2020) and regularized dual channel algorithm (RDCA) (Chaubell et al., 2020), suggested there is still uncertainty about how SMAP observations of horizontal and vertical brightness temperature can be best translated into estimates of surface properties. Although SMAP can provide spatially explicit soil moisture estimates that have been shown to be useful to understand a set of ecohydrological problems (Dadap et al., 2019; Feldman et al., 2018), the soil moisture retrievals are still subject to significant amount of uncertainty due to the imperfection of the model and the forcing datasets. It is also important to consider the how the amount of duplicate information carried within a set of observations limits the number of independent parameters to be inferred (Konings et al., 2015). Therefore, it is critical to diagnosis and quantify the causality of the uncertainty caused

by the SMAP algorithm to improve the soil moisture and vegetation opacity retrieval quality.

SMAP soil moisture products have been extensively validated against well-calibrated *in situ* soil moisture using unbiased root mean square error (ubRMSE), bias, RMSE Pearson correlation coefficients and triple collocation method at 'core' and 'sparse' validation sites (Chan et al., 2016; Chen et al., 2017; Colliander et al., 2017; Zhang et al., 2019). These validation investigations found that SMAP met the required accuracy target (ubRMSE, 0.04 m$^3$/m$^3$) on average, while there exist some locations where the performance of SMAP did not met the expected performance. All these validation studies were focused on finding the

general uncertainty of SMAP (which is the deviation of SMAP soil moisture from the *in situ* or reference soil moisture) and cannot diagnose and differentiate where the uncertainty arises. Indeed, the causality of uncertainty of SMAP soil moisture may arise from two aspects: (1) the uncertainty due to the inaccuracies from forcing the datasets and (2) the uncertainty due to poor model structure and parameterizations. In addition, the assessment metrics used in these evaluation studies are either heavily dependent on *in situ* soil moisture or additional reference datasets, which does not allow for SMAP to be validated in some

remote and inaccessible areas.

The challenges faced by previous SMAP evaluation investigations can be resolved by leveraging two information quantities : (1) Shannon's entropy (Shannon, 1948), which is the amount of information required to fully describe a random variable and (2) mutual information (Cover and Thomas, 2005), which represents the amount information of knowing one variable given

the knowledge of another or a set of random variables. Gong et al., 2013 first leveraged these information quantities to partition overall uncertainty in the hydrological modeling process into two categories: (1) random uncertainty that arises by incompleteness of exploratory variable and/or inherent stochasticity of forcing datasets, and (2) model uncertainty that is contributed by poor model parameterization or formulation. The random uncertainty is not resolvable for the given system as it is only related to the probability distributions of the forcing data itself, while the model uncertainty is reduceable by a better

model parameterization.

Given that both horizontal and vertical polarized brightness temperatures are measured by SMAP, it is unclear how each polarization contributes information to the overall performance of the DCA. Recent research on partial information decomposition has provided tremendous opportunities for understanding the nuanced interactions among different variables

and model structure. Initially proposed by Williams and Beer, 2010 and further advanced by Goodwell and Kumar, 2017, this approach has been used to understand environmental processes that link two source variables with a target variable by partitioning multivariate mutual information into unique, redundant and synergistic components. The unique information represents the amount of information shared with the target variable from each individual source variable separately (Finn and Lizier, 2018). Synergistic information is the information provided to the target while both source variables act jointly (Kunert-

Graf et al., 2020). Redundant information is the overlapping information that both source variables redundantly provide to a target (Wibral et al., 2017). Information partitioning brings new insight by unambiguously characterizing the interdependencies between source variables and a target variable without any underlying assumption (Goodwell et al., 2018).

The overall objective of this study is to demonstrate that by assessing how information flows through satellite algorithms from

raw retrievals to end user products, we can illuminate areas where improvements can be made and diagnose instances where algorithm estimates are expected to be uncertain. In this study, we focus on (1) quantifying the random uncertainty and model uncertainty in SMAP's Dual Channel Algorithm (DCA) and understand how these uncertainties are related to DCA retrieval quality; (2) exploring how the partial information components between SMAP DCA soil moisture and horizontally polarized and vertically polarized brightness temperature can be used to indicate overall DCA soil moisture retrieval performance.


## 2 Material and Methods

### 2.1 *In situ* soil moisture

The US Climate Reference Network (USCRN) is a systematic and sustained network that is operated and maintained by National Oceanic and Atmospheric Administration (NOAA) to support climate-impact research with continuous high-quality

field observed soil moisture, soil temperature and windspeed at different temporal scales (Diamond et al., 2013). The USCRN provides soil moisture observations at five different standard depth (5 cm, 10 cm 20 cm, 50 cm, and 100 cm) in 114 locations of Contiguous U.S. (CONUS) (Bell et al., 2013). These *in situ* datasets have been used for a wide variety of research such as drought evaluation and satellite soil moisture validation (Bell et al., 2015; Leeper et al., 2017). The hourly soil moisture (beta version product) datasets at the depth of 5 cm were collected from 58 (15 croplands, 32 grasslands, 5 shrublands, 2 savannas,

4 mixed) selected USCRN stations (Fig. 1 and Table S1) based on the availability of *in situ* soil moisture dataset and the data quality of SMAP pixels in the study period of March 31, 2015 to December 10, 2020.

### 2.2 SMAP Level-2 datasets

In this study, we acquired the water body corrected horizontally polarized brightness temperature ($T_{Bh}$), vertically polarized

brightness temperature ($T_{Bv}$), soil effective temperature ($T_{eff}$), DCA soil moisture and the fraction of landcover at each selected USCRN station from SMAP Level-2 Radiometer Half-Orbit 36 km EASE-Grid Soil moisture, Version 7 data product (O'Neill et. al., 2020b) in the same period as the USCRN soil moisture at every station. The extracted data series were filtered by the internal quality flags of $T_{Bh}$ ("tb_qual_flag_h"), $T_{Bv}$ ("tb_qual_flag_v") and DCA ("retrieval_qual_flag_option3") as provided in SMAP data files. We retain data points at a particular SMAP observation time when they all simultaneous pass quality

control and fall within their correspondent valid ranges (e.g., 0 ~ 330K for $T_{Bh}$ and $T_{Bv}$, 253.15K ~ 313.15K for $T_{eff}$, > 0.02m$^3$/m$^3$ for DCA soil moisture) as specified in the SMAP documentation (Chan, 2020). On average, the number of datapoints across all the sites is 1090 with a minimum of 225 and a maximum of 1652. DCA retrieves soil moisture based on the '*tau-omega*' model (Jackson et al., 1982; Mo et al., 1982), which is a well-known radiative transfer based soil moisture retrieval algorithm in the passive microwave soil moisture community. It requires the brightness temperatures as the main

inputs, soil effective temperature as an ancillary input, and is parameterized based on overlaying vegetation and soil surface information (Njoku and Entekhabi, 1996). The DCA iteratively feeds the '*tau-omega*' model with initial guesses of soil moisture and vegetation optical depth. The retrieved soil moisture is assumed to be close to the real value when the estimated brightness temperatures are similar to the satellite observed brightness temperature (Konings et al., 2017; O'Neill et. al., 2020a). Compared to the SCAs, the DCA uses a different polarization mixing factor function and different values of vegetation single

scattering albedo (O'Neill et. al., 2020a).

The SMAP fraction of landcover data field provides the fraction of top three dominate landcovers that were classified by International Geosphere – Biosphere Programme (IGBP) ecosystem surface classification scheme at each pixel (Chan, 2020). The IGBP classified land surface into water, evergreen needleleaf forest, evergreen broadleaf forest, deciduous needleleaf forest, deciduous broadleaf forest, mixed forest, closed shrublands, open shrublands, woody savannas, savannas, grasslands, permanent wetlands, croplands, urban and built-up, croplands/natural vegetation mosaics, snow and ice, barren (Seitzinger et al., 2015). In this study, the landcover of the study site was classified as the most dominate landcover if the fraction of the most dominate landcover was greater than 50%. Otherwise, the landcover of the study site is classified as the "mixed" landcover. Furthermore, the study sites that are dominated by woody savanna were classified as savannas, by closed/open shrublands were classified as shrublands, by cropland/natural vegetation mosaics were classified as croplands. Sites meeting specified data requirements and their associated landcover classification are shown in Figure 1. Additionally, the 500m leaf area index (LAI) of each site was obtained from NASAs Moderate Resolution Imaging Spectrometer (MODIS) mission (Myneni et. al., 2015; ORNL DAAC, 2018) and averaged in time. Within each site the mean and standard deviation of LAI of all pixels within each SMAP pixel was calculated as a measure of vegetation biomass and variability.

## 2.3 Information – based uncertainty partitioning

The fundamental quantity of information theory is Shannon's entropy (Shannon, 1948), which represents the amount of information required to fully describe a random variable (Cover and Thomas, 2005). Shannon's entropy is the basic building block of computing mutual information and the informational uncertainties. The entropy of a single random variable is defined as

$$H(\text{X}) = -\sum_{x \in X} p(x) log_2 \, p(x), \tag{1}$$

where $p(x)$ is the probability mass function of random variable X. The estimation of $p(x)$ often involves discretizing the values of X into a set of bins and then the $p(x)$ of a specific bin is computed by dividing the total number of datapoints within a specific bin by the total of number of data points of X. The number of bins in this study is estimated by Freedman-Diaconis binning method (Freedman and Diaconis, 1981). The entropy calculated by eq. (1) is in unit of bits.

Previous study has indicated that this method (eq. (1)) may underestimate the true entropy (Paninski, 2003). Therefore, we leveraged the simple Miller-Madow corrected entropy estimator (Zhang and Grabchak, 2013) and we also normalization the entropy to remove the bias that may cause by the heterogeneity in length of available datasets across all stations. We acknowledge that there exist several entropies estimation methods. However, we select the Miller-Madow correction based on its simplicity and effectiveness. The corrected and normalized entropy is then expressed as

$$H_{CN}(\text{X}) = \frac{H(\text{X}) + \frac{K-1}{2n}}{log_2 \, n}, \tag{2}$$

where $H_{CN}(\text{X})$ is the Miller-Madow corrected and normalized entropy of random variable X (hereafter entropy), $H(\text{X})$ is the uncorrected entropy from eq. (1), $n$ is the number of data points of X , $K$ is the number of non-zero probabilities (bins contains more than one data point) based on the fixed binned method (Freedman and Diaconis, 1981). In this study, all entropies of

single random variables in the later equations (i.e., $H_{CN}(T_{Bh})$, $H_{CN}(T_{Bv})$, $H_{CN}(in\ situ)$ etc.) are computed using the combination of eq. (1) and eq. (2) with the replacement of $p(\bullet)$ by their individual probability mass functions.

The joint entropy (Cover and Thomas, 2005) is a critical intermediate information quantity to calculate these informational uncertainties. It represents the amount of information required to describe a set of random variables. The joint entropy for two random variables is defined as

$$H(X, Y) = -\sum_{x \in X} \sum_{y \in Y} p(x,y) log_2\ p(x,y), \tag{3}$$

where $p(x, y)$ is the joint probability mass function associated with X and Y that is estimated by the same method mentioned above. The same normalization and correction method of eq. (2) is applied to joint entropy of eq. (3). The entropy after the correction and normalization is formulated as

$$H_{CN}(X, Y) = \frac{H(X,Y) + \frac{K-1}{2n}}{log_2\ n}, \tag{4}$$

where $H_{CN}(X, Y)$ is the corrected and normalized joint entropy of random variable associated with $\{X, Y\}$, $H(X, Y)$ is the uncorrected and unnormalized entropy from eq. (3), $n$ is the number of data points that were used to calculate the normalized joint entropy (hereafter joint entropy), $K$ is the number of non-zero joint probabilities based on the Freeman and Diaconis method (Freedman and Diaconis, 1981). All the joint entropies that are associated with two or more random variables in the later equations (i.e., $H_{CN}(in\ situ,\ DCA)$, $H_{CN}(T_{Bh},\ T_{Bv},\ DCA)$, $H_{CN}(T_{Bh},\ T_{Bv},\ T_{eff},\ in\ situ)$ etc.) are computed using the combination of eq. (3) and eq. (4) with the replacement of $p(\bullet)$ by their joint probability mass functions, respectively.

Generally, modeling efforts are focused on capturing the information of a random variable of interest via other explanatory variables through some physically- or empirically- based models. However, most of models being constructed of natural processes are not perfect, and the model outputs are often not capable of capturing the exact relationship between the available input variables and the variable of interest (Gong et al., 2013). There exists a maximum achievable performance of a model that describes the variable of interest the best for a particular system given the available datasets (Gong et al., 2013); yet the detailed structure of this model is often unknown. Mutual information (Cover and Thomas, 2005), for instance $I(A; B)$, is a measure of the amount information due to the knowledge of knowing either random variable A or B in the function $I(\bullet;\bullet)$. Mutual information between model inputs and *in situ* observations of model output (hereafter *in situ* observations) can be used as a useful and effective measure of best achievable performance model because it links the model inputs and *in situ* observations only through the joint and marginal probability mass functions that do not involve any priori model assumptions (Gong et al., 2013).

The mutual information can be defined based on entropy and joint entropy (Cover and Thomas, 2005). The mutual information between $T_{Bh}$ and DCA, and the mutual information between $T_{Bv}$ and DCA, are computed as

$$I(\text{T}_{\text{Bh}}; \text{DCA}) = H_{CN}(\text{T}_{\text{Bh}}) + H_{CN}(\text{DCA}) - H_{CN}(\text{T}_{\text{Bh}}, \text{DCA}) \tag{5}$$

and

$$I(\text{T}_{\text{Bv}}; \text{DCA}) = H_{CN}(\text{T}_{\text{Bv}}) + H_{CN}(\text{DCA}) - H_{CN}(\text{T}_{\text{Bv}}, \text{DCA}). \tag{6}$$

The mutual information between *in situ* and DCA soil moisture is computed as

$$I(\text{DCA}; in\ situ) = H_{CN}(\text{DCA}) + H_{CN}(in\ situ) - H_{CN}(\text{DCA}, in\ situ). \tag{7}$$

The mutual information between DCA and *in situ* soil moisture is calculated as

$$I(\text{T}_{\text{Bh}}, \text{T}_{\text{Bv}}; \text{DCA}) = H_{CN}(\text{T}_{\text{Bh}}, \text{T}_{\text{Bv}}) + H_{CN}(\text{DCA}) - H_{CN}(\text{T}_{\text{Bh}}, \text{T}_{\text{Bv}}, \text{DCA}). \tag{8}$$

The mutual information between $\text{T}_{\text{Bh}}$, $\text{T}_{\text{Bv}}$, $\text{T}_{\text{eff}}$ and *in situ* soil moisture is computed as:

$$I(\text{T}_{\text{Bh}}, \text{T}_{\text{Bv}}, \text{T}_{\text{eff}}; in\ situ) = H_{CN}(\text{T}_{\text{Bh}}, \text{T}_{\text{Bv}}, \text{T}_{\text{eff}}) + H_{CN}(in\ situ) - H_{CN}(\text{T}_{\text{Bh}}, \text{T}_{\text{Bv}}, \text{T}_{\text{eff}}, in\ situ). \tag{9}$$

We adopted the information uncertainty analysis by Gong et al., 2013 and applied it to SMAP DCA. For a given system in which the inputs and output are linked via mathematical functions, the mutual information between model output and *in situ* observation can never exceed the entropy of the *in situ* observations. Conceptually, the entropies of model output and *in situ* observations can be considered as two circles (of equal or unequal sizes) and the mutual information between model output and *in situ* observation can be viewed as the overlapping area of these two circles (Uda, 2020). Therefore, the maximum mutual information shared between model output and *in situ* is the minimum of the entropy of model output and *in situ* observations, i.e: $I(\text{DCA}, in\ situ) \leq \min[H_{CN}(\text{DCA}), H_{CN}(in\ situ)]$. Intuitively, the overlapping area of two circles cannot be larger that of the smaller circle. Because we are focused on representing the observed soil condition, the information gap between *in situ* observations, $H_{CN}(in\ situ)$, and the mutual information shared between *in situ* observations and model output, $I(\text{DCA}, in\ situ)$, is defined as informational total uncertainty ($I_{Tot}$). This quantity describes how much of the information within *in situ* observations, as measured by $H_{CN}(in\ situ)$, is not captured by the estimator, as measured by $I(\text{DCA}, in\ situ)$. The mutual information between the *in situ* observations and the available explanatory variables is also always smaller than the entropy of *in situ* observations. This information gap, defined as informational random uncertainty ($I_{Rnd}$), is caused by the existence of inherent data uncertainty of the explanatory variables and a lack of complete explanatory variables to fully capture the information in the *in situ* observations (Gong et al., 2013). Furthermore, the mutual information between model inputs and *in situ* observations should equal to the mutual information between *in situ* observations and model output if the model hypothesis completely captures the true relationship between model inputs and *in situ* observations. However, it's commonly stated that "*All models are wrong*" (Box, 1976) and model assumptions typically cannot fully express the true relationship between the explanatory variables and *in situ* observations. Hence, the mutual information between model output and *in situ* observation is

expected to be smaller than the mutual information between model inputs and *in situ* observations (Gong et al., 2013). This information gap, defined as informational model uncertainty ($I_{Mod}$) is induced by poor model assumption, formulations, and/or inappropriate model parameterizations. Therefore, the informational total uncertainty ($I_{Tot}$) is the sum of the informational random uncertainty and informational model uncertainty come naturally given the explicitly definition of these informational uncertainties.

In this study, the explanatory variables of DCA are $T_{Bh}$, $T_{Bv}$ and the $T_{eff}$. The *in situ* observation and model output are *in situ* USCRN soil moisture and DCA soil moisture, respectively. Leveraging eq. (7) and eq. (9), the DCA informational random uncertainty ($I_{Rnd}$), DCA informational model uncertainty ($I_{Mod}$), and DCA total informational uncertainty ($I_{Tot}$) calculated are calculated as:

$$I_{Rnd} = H_{CN}(in\ situ) - I(T_{Bh}, T_{Bv}, T_{eff};\ in\ situ), \tag{10}$$

$$I_{Mod} = I(T_{Bh}, T_{Bv}, T_{eff};\ in\ situ) - I(DCA;\ in\ situ), \tag{11}$$

and

$$I_{Tot} = H_{CN}(in\ situ) - I(DCA;\ in\ situ) = I_{Rnd} + I_{Mod}. \tag{12}$$

## 2.4 Partial information decomposition

The distinct informational contributions of $T_{Bh}$ and $T_{Bv}$ to the DCA soil moisture are assessed through a decomposition of the mutual information. This method partitions multivariate mutual information to unique, redundant and synergistic components (Williams and Beer, 2010). The decomposed information components on the DCA model inputs and outputs are expected to indicative of informational flow as model inputs are translated to end user products, and these components may have potential for evaluating model performance. The partial information decomposition of $I(T_{Bh}, T_{Bv};\ DCA)$ can be expressed as

$$I(T_{Bh}, T_{Bv};\ DCA) = U_h(T_{Bh};\ DCA) + U_v(T_{Bv};\ DCA) + R(T_{Bh}, T_{Bv};\ DCA) + S(T_{Bh}, T_{Bv};\ DCA), \tag{13}$$

where $U_h$ and $U_v$ are unique information of $T_{Bh}$ and $T_{Bv}$ shared with DCA, respectively. $S$ and $R$ are the synergistic information and redundant information that $T_{Bh}$ and $T_{Bv}$ shared with DCA estimates, respectively. All the decomposed components are non-negative real values (Williams and Beer, 2010).

The mutual information between $T_{Bh}$ and DCA and mutual information between $T_{Bv}$ and DCA are formulated as

$$I(T_{Bh};\ DCA) = U_h(T_{Bh}; DCA) + R(T_{Bh}, T_{Bv};\ DCA) \tag{14}$$

and

$$I(T_{Bv};\ DCA) = U_v(T_{Bv};\ DCA) + R(T_{Bh}, T_{Bv};\ DCA). \tag{15}$$

In this approach, $U_h$, $U_v$, $S$ and $R$ are unknowns in the systems of equations (13) - (15). Goodwell and Kumar (2017) showed that the $R$ can be formulated as

$$R = R_{\min} + I_s*(R_{MMI} - R_{\min}),$$ (16)

where

$$I_s = \frac{I(T_{Bh}; T_{Bv})}{\min\{H_{CN}(T_{Bh}); H_{CN}(T_{Bv})\}},$$ (17)

$$R_{MMI} = \min[I(T_{Bh}; DCA), I(T_{Bv}; DCA)]$$ (18)

and

$$R_{\min} = \max(0, -II)$$ (19)

The $II$ is the interaction information of $T_{Bh}$, $T_{Bv}$, DCA and can be computed as:

$$II = I(T_{Bh}; DCA| T_{Bv}) - I(T_{Bh}; DCA) = H_{CN}(T_{Bh}, DCA) + H_{CN}(T_{Bv}, DCA) +$$
$$H_{CN}(T_{Bh}, T_{Bv}) - H_{CN}(T_{Bh}) - H_{CN}(T_{Bv}) - H_{CN}(DCA) - H_{CN}(T_{Bh}, T_{Bv}, DCA)$$ (20)

It is important to note that we used the point based *in situ* soil moisture as the ground truth in this analysis. Due to coarse spatial resolution of SMAP products, we acknowledge that *in situ* soil moisture may not be able to represent the spatial averaged soil moisture well. Although the nominal sensing depth of L-band SMAP soil moisture is 5 cm, the penetration depth was found to be even shallower in wetter regions (Shellito et al., 2016). In fact, the L-band sensing depth was found to as little as ~1cm (Jackson et al., 2012) and was found to vary with surface soil moisture conditions (Escorihuela et al., 2010; Raju et al., 1995). The heterogeneity in each pixel relative to the *in situ* observations together with the sensing depth disparity may influence the results of this study and can bias the estimation of informational uncertainties. We also acknowledge the existence of upscaling methods for matching the *in situ* soil moisture to satellite footprint (Crow et al., 2012). However, most of upscaling methods are achieved under the assistance of additional reference soil moisture datasets. This process introduces additional pieces of information in the DCA system making the separation of the uncertainty induced by the upscaling algorithm or additional dataset from other informational uncertainties much harder. Additionally, we used the hourly *in situ* data to best match the SMAP DCA soil moisture retrievals in time (within an hour). Based on current technologies, it is difficult to find a reference dataset with high frequency in time domain and good spatial coverage. Here we consider the informational uncertainty caused by the spatial mismatch and sensing depth mismatch between *in situ* and DCA soil moisture as part of the informational random uncertainty ($I_{Rnd}$) because the DCA is essentially a mathematical function and does not inherently require the inputs to be at a specific resolution. The spatial resolution is often the inherent attribute of the data. The reader should also keep these in mind while interpreting and adopting the results in this study.

## 3 Results
### 3.1 Information quantities and system informational uncertainties

The estimated entropies across all the study sites are shown in Figure 2 while the mutual information quantities are shown in Figure 3. The brightness temperature entropies, $H_{CN}(T_{Bh})$ and $H_{CN}(T_{Bv})$, generally follow the same pattern across sites with both having an average value of 0.37. Although the patterns of $H_{CN}(T_{Bh})$ and $H_{CN}(T_{Bv})$ are similar, the $H_{CN}(T_{Bh})$ is slightly more variable than $H_{CN}(T_{Bv})$ with the coefficients of variation (CV) being 0.053 and 0.046, respectively. $H_{CN}(T_{eff})$ shares the same average with $H_{CN}(T_{Bh})$ and $H_{CN}(T_{Bv})$, whereas the pattern of $H_{CN}(T_{eff})$ is quite different (Fig. 2). On average, the $H_{CN}(in\ situ)$ is 0.35, while $H_{CN}(DCA)$ is 0.38. In general, $H_{CN}(DCA)$ follows the pattern of $H_{CN}(in\ situ)$ with the CV of $H_{CN}(DCA)$ (0.064) being smaller than the CV of $H_{CN}(in\ situ)$ (0.081).

As shown in Figure 4a, the entropies of the retrieved brightness temperatures and DCA model output, $H_{CN}(T_{Bh})$, $H_{CN}(T_{Bv})$ and $H_{CN}(DCA)$, are significantly correlated with the entropy of $in\ situ$ observations, $H_{CN}(in\ situ)$, while no significant correlation is found between $H_{CN}(in\ situ)$ and $H_{CN}(T_{eff})$. The $H_{CN}(DCA)$ has the strongest correlation strength with $H_{CN}(in\ situ)$ compared with other entropy quantities (Fig. 4a). As expected, the mutual information quantities (Fig. 3) are shown to be generally smaller than the entropy quantities (Fig. 2). On average, $I(T_{Bh},T_{Bv};\ DCA)$ is 0.13, while the $I(DCA;\ in\ situ)$ and $I(T_{Bh},T_{Bv},\ T_{eff};\ in\ situ)$ are 0.07 and 0.17 (Fig. 3), respectively. $I(T_{Bh},T_{Bv},\ T_{eff};\ in\ situ)$ and $I(T_{Bh},T_{Bv};\ DCA)$ are significantly correlated (0.59 and -0.31) with $H_{CN}(in\ situ)$, while no significant correlation is found for $I(DCA;\ in\ situ)$ and $H_{CN}(in\ situ)$ (Fig. 4b).

It is noticeable that there exists a large information gap between $H_{CN}(in\ situ)$ in Figure 2 and $I(T_{Bh},T_{Bv},\ T_{eff};\ in\ situ)$ and $I(T_{Bh},T_{Bv},\ T_{eff};\ in\ situ)$ and $I(DCA;\ in\ situ)$ in Figure 3. These information gaps confirm the existence of informational random uncertainty ($I_{Rnd}$) and informational model uncertainty ($I_{Mod}$) in the SMAP DCA system. When calculating informational quantities on a site-by-site basis and then averaging, the SMAP DCA explains 20% of the $H_{CN}(in\ situ)$ leaving 80% of the $H_{CN}(in\ situ)$ that is unexplained (Table 1) as informational total uncertainty ($I_{Tot}$). 35% of the $I_{Tot}$ is caused by $I_{Mod}$, while the rest is induced by $I_{Rnd}$. The information uncertainties vary slightly across different landcovers. On average across sites, the SMAP DCA system is capable of capturing more information of $H_{CN}(in\ situ)$ at croplands and savannas (Table 1). Shrublands have largest absolute $I_{Rnd}$ (0.21) than other landcovers, while savannas have the largest proportion of $I_{Rnd}$ to $I_{Tot}$ (Table 1). $I_{Mod}$ in absolute value is greater in shrublands, grasslands, and croplands with grasslands have the largest proportion of $I_{Mod}$ to $I_{Tot}$ (Table 1). When lumping all the datasets together and recalculating informational quantities, we observe that SMAP DCA captures 10% of the information in the $in\ situ$ soil moisture and the proportion of $I_{Mod}$ to $I_{Tot}$ is larger.

The relationship between different informational uncertainties and the Pearson correlation coefficients between $in\ situ$ soil moisture and SMAP DCA soil moisture, a commonly adopted relative model evaluation metric in SMAP studies (Chan et al., 2016; Colliander et al., 2017), was evaluated. The $I_{Tot}$, $I_{Mod}$ and $I_{Rnd}$ are shown to be related how well the SMAP DCA soil moisture is correlated with $in\ situ$ soil moisture (Fig. 5). $I_{Tot}$ is found to be negatively correlated ($r = -0.69$, Fig. 5a) with the Pearson correlation between $in\ situ$ soil moisture and SMAP DCA soil moisture. Similarly, $I_{Mod}$ and $I_{Rnd}$ are also shown to be negatively (-0.59 and -0.34 respectively) related to the Pearson correlation between $in\ situ$ soil moisture and SMAP DCA soil moisture with $I_{Mod}$ being more influential than $I_{Rnd}$ (Fig. 5b and 5c). These negative relationships are consistent with general expectations since SMAP tends to capture more information about the $in\ situ$ soil moisture (i.e. lower $I_{Tot}$, $I_{Mod}$ and $I_{Rnd}$) when it retrieves high quality datasets (higher correlation between $in\ situ$ soil moisture and SMAP DCA soil moisture).

## 3.2 Partial information decomposition of DCA

The partial information decompositions were assessed on a site basis and are shown in Figure 6. The fractional contribution of each component to that site's mutual information between brightness temperatures and DCA estimates, $I(T_{Bh},T_{Bv}; DCA)$, was also calculated and are given in Table 2. Generally, the majority of $I(T_{Bh},T_{Bv}; DCA)$ is redundantly ($R$) shared by $T_{Bh}$ and $T_{Bv}$, which is about 0.08 (58% of $I(T_{Bh},T_{Bv}; DCA)$) on average (Table 2). The mean values of unique information of $T_{Bh}$ ($U_h$) and synergistic information ($S$) of $T_{Bh}$ and $T_{Bv}$ are 0.024 (18% of $I(T_{Bh},T_{Bv}; DCA)$) and 0.018 (14% of $I(T_{Bh},T_{Bv}; DCA)$), respectively (Table 2). Compared to other decomposed information components, $U_v$ is the smallest with its mean being 0.013 (10% of $I(T_{Bh},T_{Bv}; DCA)$). Savannas have the highest absolute and fraction of $R$ (0.101 ,74% of $I(T_{Bh},T_{Bv}; DCA)$) (Table 2). In general, the DCA system is mainly dominated by $R$ as indicated by both site wise decomposition and when lumping all datasets together (45% of $I(T_{Bh},T_{Bv}; DCA)$) and $S$ is consistently the lowest (Table 2).

Through this analysis, it is shown (Fig. 7) that there are strong relationships between SMAP DCA retrieval quality and decomposed information components. In general, the correlation strength between DCA and *in situ* soil moisture is higher when $U_h$, $U_v$ and $S$ are low and $R$ is high. This is demonstrated by a significant correlation of these components with the Pearson correlation between *in situ* and DCA soil moisture. The negative relationship between increasing $S$ and decreasing DCA quantity is strongest of the decomposed components, though the positive relationship between increasing $R$ and decreasing DCA is of similar correlation strength. This indicates that $R$ or $S$ contains useful information about DCA soil moisture quality.

## 4 Discussion

### 4.1 DCA informational uncertainties

The first objective of this study is to leverage information theory to quantitatively decompose the informational total uncertainty into informational random uncertainty and informational model uncertainty in the DCA as an approach to understand where retrieval uncertainties arise. This information theory approach can provide new insight to SMAP modeling diagnosis. It offers an opportunity of partitioning the total informational uncertainty in the DCA into the uncertainty due to the input datasets and the uncertainty due to model structure and model parameterizations. This partition process cannot be achieved by leveraging the common DCA assessment metrics (Chan et al., 2016) (e.g., Pearson correlation, ubRMSE) that only involve the DCA soil moisture and *in situ* soil moisture.

The DCA model structure is inherently a hypothesis that relates the input datasets to soil moisture based on prior physical knowledge. The DCA is thus a procedure of processing the input dataset into estimates soil moisture. Thus, models, even those that perform the best, can only reduce the available information in its inputs and are not capable of adding new information about the "true" soil moisture. Hence, there is no possibility of building a model that is better than the one with the best achievable performance of the input data themselves (yet even achieving this theoretically limit is nearly impossible) (Gong et al., 2013). If, however, more freedom on available datasets to incorporate is given, it is possible to build models that outperform the best achievable model performance by adding new explanatory variables which may lead to a family of models that have completely different model structure. Based on Table 1, we find that the DCA has more informational uncertainty in

shrublands than grasslands and croplands. This might be due to stronger variability in vegetation in for shrublands while grasslands and croplands tend to be more uniform and homogeneous. It is worth noting that these finding are based on averaging our studied sites within different landcover categories, and results may be different while comparing two specific sites from different landcovers. In addition, we find the proportion of informational uncertainty increases as the data is lumped together relative to averaging these statics calculated on a site-by-site basis (Table 1). Treating all the surfaces together as a whole does not reduce the informational total uncertainty because the lumping process contains both "high quality" and "low quality" (as assessed by the Pearson correlation between *in situ* and DCA soil moisture) datasets. The uncertainties in these datasets may accumulate while lumping them together and result in an increase in total informational uncertainty.

The fraction that informational random uncertainty contributes to the informational total uncertainty is quite significant (65% on average) in this study. The informational random uncertainty in the system may arises from the inherent error due to calibration of $T_{Bh}$ and $T_{Bv}$ (Al-Yaari et al., 2017), the mismatch in the scale of observations, and the presence water bodies (Ye et al., 2015). If poorly calibrated, the soil moisture estimations can be exacerbated due to the error propagation that hinders the correct information being expressed. Furthermore, SMAP attempts to $T_{eff}$ to capture both soil and canopy temperature because the differences between canopy and soil temperature are minimized in the morning and afternoon orbits. The $T_{eff}$ is computed based on a model that uses the information from average soil temperature of first layer and second layer and interpolated in time in order to match SMAP observations (O'Neill et. al., 2020a). These interpolation and modeling processes may produce erroneous $T_{eff}$ dataset and hence contribute the informational random uncertainty of DCA. Therefore, a better and robust calibration strategy of $T_{Bh}$ and $T_{Bv}$ to the presence of water bodies and a comprehensive assessment of $T_{eff}$ may be needed to reduce some of the information random uncertainty.

Informational model uncertainty contributes an unneglectable portion to the informational total uncertainty (35% on average). This model uncertainty may arise from poor model parameterizations, which may vary with site soil moisture dynamics ($H_{CN}$(*in situ*)). As shown in figure 4b, the $I(T_{Bh},T_{Bv}, T_{eff}; in situ)$ increases as the *in situ* soil moisture is more dynamic as reflected by high values of $H_{CN}(T_{Bh})$ and $H_{CN}(T_{Bv})$. The raw observations ($T_{Bv}$, $T_{Bh}$, and $T_{eff}$) provide more available information to the system, whereas such information is not properly captured by the algorithm as reflected by low correlation strength between $H_{CN}$(*in situ*) and $I$(DCA; *in situ*). Therefore, it is more likely to observe large information model uncertainty where the soil moisture is more dynamic, which may cause a low efficiency of DCA to correctly transmit the available information. It is known that DCA is parameterized with a set of surface and vegetation parameters such as vegetation single scattering albedo ($\omega$), surface height standard deviation ($s$), etc. These parameter values are landcover dependent and are derived from past studies as well as prior experience and some information discussions with experts, all of which could be biased and inaccurate (O'Neill et. al., 2020a). These parameter values also are not differentiated by landcover microwave polarization directions and were assumed to be constant in time. It is possible that these parameters (such as $\omega$) vary in time (Konings et al., 2017) and shift during senescence or harvesting seasons. It is observed that the proportion of the informational model uncertainty is slightly smaller in shrublands (Table 1) (here we do not include savannas in the discussion since this landcover only have 2 sites), while these proportions are larger in croplands and grasslands (Table 1). This might because the model parameterizations are more reasonable in shrublands than other landcovers. In addition, croplands and grasslands may have seasonal harvesting

and therefore may more subject to changes in these values, while shrublands may not. Additionally, when averaging informational values site-by-site, the informational random uncertainty is a larger fraction of the total uncertainty, whereas when all data are lumped together, the informational model uncertainty is a larger fraction (Table 1). DCA parameters are different with respect to each landcover, and the biases induced by these parameters at each site may accumulate through the system resulting a dominance in informational model uncertainty over informational random uncertainty when all sites are lumped together.

To summarize, this is the first attempt of leveraging mutual information approach to analyze the uncertainty components in microwave remote sensing models. The results of this study can be further used as guidance in assessing of SMAP algorithm and can quantitively identify where information lost in the process of SMAP soil moisture modeling. More broadly, this study, though focused on SMAP, can be transferred and extended to analyze other remote sensing algorithms. Over many decades, a lot of effort, resources, and time have been devoted to the launch numerous of satellite missions to retrieve the key environmental variables such as evapotranspiration and vegetation biomass (Dubayah et al., 2020; Hulley et al., 2017). Performing such analysis on these retrieval algorithms is expected to be beneficial to understanding the informational flow in these algorithms and may provide insights to further improve the data retrieval accuracy as well as making maximum use of data collected at greater expense.

## 4.2 Model evaluation from another perspective

The second objective of this study was to demonstrate that the partitioned information components contain useful information about DCA model performance that does not depend on *in situ* soil moisture and other ancillary datasets. We find a strong linear relationship between redundant ($R$) and synergistic ($S$) information of the polarized brightness temperatures and Pearson correlation between DCA and *in situ* soil moisture. In general, it is more likely to observe higher $R$ and lower $S$ (and $U_h$ and $U_v$) in the less woody landcovers such as croplands and grasslands, where the range of brightness temperature may possibly be greater. These information components were found to be marginally correlated with factors such as vegetation density (the Pearson correlation of average LAI with $R, S, U_h, U_v$ are 0.23, -0.38, -0.54, and -0.19 respectively) and vegetation heterogeneity (the Pearson correlation of LAI standard deviation with $R, S, U_h, U_v$ are 0.22, -0.39, -0.52, and -0.22 respectively). Additionally, these informational components were also found to be correlated with the mutual information shared between brightness temperatures and DCA estimates (the Pearson correlation of $I(\text{T}_{Bh}, \text{T}_{Bv}; \text{DCA})$ with $R, S, U_h, U_v$ are 0.6, -0.28, 0.22, and -0.16 respectively), the informational total uncertainty (the Pearson correlation of $I_{Tot}$ with $R, S, U_h, U_v$ are -0.76, 0.62, 0.56, and 0.68 respectively), informational random uncertainty (the Pearson correlation of $I_{Rnd}$ with $R, S, U_h, U_v$ are -0.42 , 0.29, 0.05, and 0.15 respectively), and informational model uncertainty (the Pearson correlation of $I_{Mod}$ with $R, S, U_h, U_v$ are -0.63, 0.56, 0.66, and 0.75 respectively). This indicates that these informational components in the DCA system are not only physically driven by both vegetation density and heterogeneity but also other factors such as how algorithm processes the information from $\text{T}_{Bh}$ and $\text{T}_{Bv}$ to produce the DCA outputs. It is more likely to observe higher $R$ and lower $S$ in locations where vegetation is denser and more heterogeneous, yet the correlation of these variables with model quality (0.47 for mean LAI and 0.42 for the standard deviation of LAI) are weaker than the correlations found between $R$ and $S$ and model quality shown in Figure 7. The $R$ and $S$ metric in this study can thus not only integrate information about how the surface vegetation density and heterogeneity

influence the algorithm performance but provided insight into how effectively DCA algorithm uses the information from $T_{Bh}$ and $T_{Bv}$.

Compared with other ancillary and *in situ* independent metrics such as correlation strength between Pearson correlation of $T_{Bh}$ with $T_{Bv}$ and the Pearson correlation between *in situ* and DCA soil moisture (0.67), the correlation strength of $S$ and $R$ with Pearson correlation of *in situ* and DCA soil moisture are tighter (0.79 and -0.82 for $R$ and $S$). This suggests the complex non-linear relationship between of $T_{Bh}$, $T_{Bv}$ with DCA soil moisture is better captured by $R$ and $S$ as compared to the direct correlation between the two brightness temperatures themselves. Given the strength of this relationship, the $R$ and $S$ holds the potential to be used as a DCA evaluation metric that does not depend on *in situ* measurement and ancillary dataset. It is also useful for SMAP DCA soil moisture users to have a rough estimation of how high the quality (as characterized as the correlation strength between DCA and *in situ*) of the obtained DCA soil moisture without actually knowing the *in situ* soil moisture. However, this depends on specific user requirements for data quality. In general, the DCA soil moisture tends to be in high end in term retrieval quality (~ 0.75 in Pearson correlation) when the $R$ is greater 0.1 or $S$ is smaller than 0.015. It is important to note that the decomposed information components are dependent on the DCA parameterizations (e.g., $\omega$, $h$. etc.) that may influence how the $T_{Bh}$ and $T_{Bv}$ are probabilistically linked with the DCA and hence may alter the partitioned information components.

## 4.3 Approach Limitations

While we expect that this approach can be generalized to analyze other remote sensing models, it may be difficult to compute the joint probability density functions for models with high-dimensional inputs. Difficulty in determining the joint probability density functions hinders the estimation of high dimensional joint entropy and mutual information components, and these are still open questions in the field of information theory. Although there exist serval data dimension reduction techniques, these dimension reduction techniques are mostly based some assumptions (Xu et al., 2019). In practice, most of the systems with high dimension inputs tend to be complex. Therefore, there is a strong risk of introducing additional uncertainty if one chooses an inappropriate technique.

It is important to understand that SMAP DCA system retrieves soil moisture with the help of vegetation water content climatology derived from the MODIS NDVI data stream. This is specified as a set value for each location and day of year combination and is used to estimate the initial guess for the unknown vegetation optical depth (O'Neill et. al., 2020a). The reader should keep in mind that this study considers such data as a dynamic time-varying parameter and it is not treated as a data input in this study. Adding NDVI as a data input would result in $I(T_{Bh}, T_{Bv}, T_{eff}, NDVI; \textit{in situ})$ being larger than or equal to $I(T_{Bh}, T_{Bv}, T_{eff}; \textit{in situ})$ in the calculation of $I_{Rnd}$, and therefore $I_{Rnd}$ would decrease. Since, $I_{Tot}$ only considers DCA output and *in situ* data it is not altered by adding dynamic parameters and $I_{Mod}$ would therefore increase. Thus, consideration of additional dynamic parameters in this informational assessment would serve to shift uncertainties from those attributed to the input data themselves to uncertainties attributed to the model structure and parameterizations.

This study was conducted only at locations where *in situ* soil moisture is readily available. It could be an interesting topic to

explore if, and how, information-based uncertainty analysis can be applied in the locations without *in situ* soil moisture measurements. We would expect the informational uncertainty analysis to provide the estimates of random and model uncertainties. The best performance we can expect from this current uncertainty analysis is to use all the available datasets we have; yet we believe that uncertainty estimations of this approach should be stabilized given adequate representative locations and data records.

## 5 Conclusions

This study differentiates and quantifies the uncertainty sources in the SMAP DCA using information theory. We found that on average DCA soil moisture explains 20% of the information in the *in situ* soil moisture leaving 80% unexplained. Among the unexplained information, 65% is informational random uncertainty that is caused by the inherent stochasticity of the explanatory variables of SMAP DCA and a lack of additional explanatory variables in the system, while the rest of the informational uncertainty is caused by inappropriateness of the assumption of DCA model structure and parameterizations. We show that informational random uncertainty contributes a larger proportion of the informational total uncertainty across different landcovers. However, the informational model uncertainty contributes more to total uncertainty when lumping all the datasets together. The performance of SMAP DCA is negatively correlated to all the information uncertainties, with the informational model uncertainty being more reflective of overall SMAP DCA retrieval quality than the informational random uncertainty.

The decomposition of the mutual information has shown that all decomposed components are significantly related to the Pearson correlation between *in situ* and DCA soil moisture, with the redundant and synergistic information being the strongest. Good DCA model performance (as measured by Pearson correlation between *in situ* and DCA soil moisture) is more likely to be found in locations where the redundant information of brightness temperatures shared with DCA soil moisture is high and is more dominant relative to other components. The informational uncertainty decomposition analysis opens a new window for SMAP algorithm uncertainty diagnosis. SMAP DCA users may examine to the $R$ and $S$ components to have an approximate estimation of the soil moisture data quality obtained when no *in situ* soil moisture is readily available.

## Code availability

The code regarding the SMAP dataset time series, mutual information and partial information decomposition calculation can obtained from https://github.com/libonancaesar/HESS_Information_Uncertainty

## Data availability

SMAP L2 Radiometer Half-Orbit 36 km EASE-Grid Soil Moisture, Version 7 is acquired from US National Snow and Ice Data Center (https://nsidc.org/data/smap). The *in situ* soil moisture is accessible through U.S. Climate Reference Network (https://www.ncdc.noaa.gov/crn/). The leaf area index dataset can be accessed through Oak Ridge National Laboratory Distributed Active Archive Center (https://modis.ornl.gov/globalsubset/).

## Author contribution

***Bonan Li***: conceptualization; data acquisition; formal analysis; methodology; original draft writing and editing; ***Stephen P. Good***: conceptualization; methodology; draft writing, editing and revisions; supervision.

## Competing interests

The authors declare no conflicts of interest.


## Acknowledgments

This project was supported by The National Aeronautics and Space Administration under grant NNX16AN13G.

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

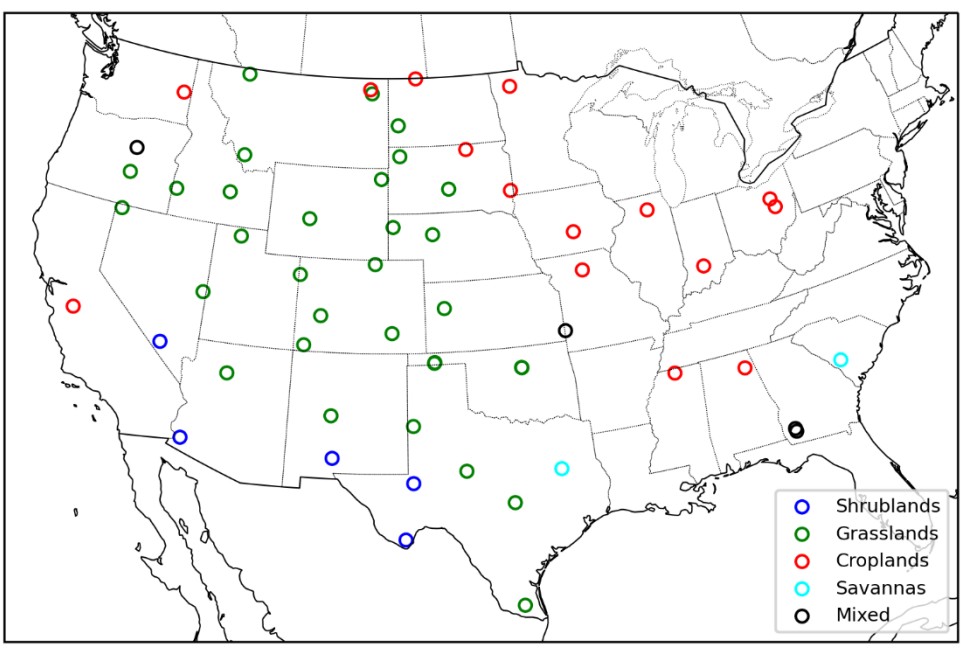

**Figure 1** Spatial distribution of selected USCRN stations classified by landcovers.


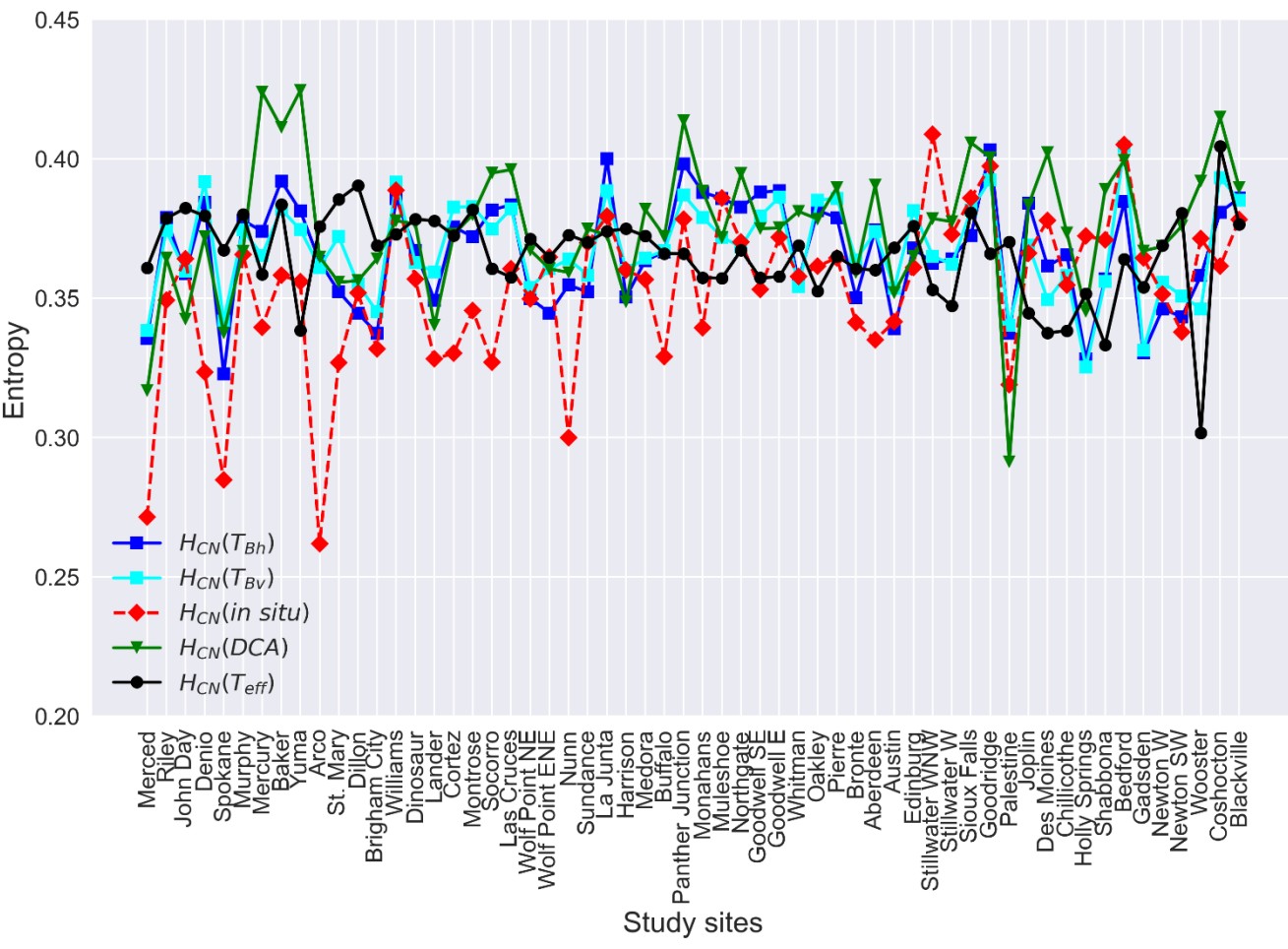

**Figure 2** Entropies of horizontally polarized brightness temperature (T$_{Bh}$), vertically polarized brightness temperature (T$_{Bv}$), *in situ* soil moisture, DCA soil moisture, and soil effective temperature (T$_{eff}$) across the study sites. The sites are ordered by longitude (West to East).

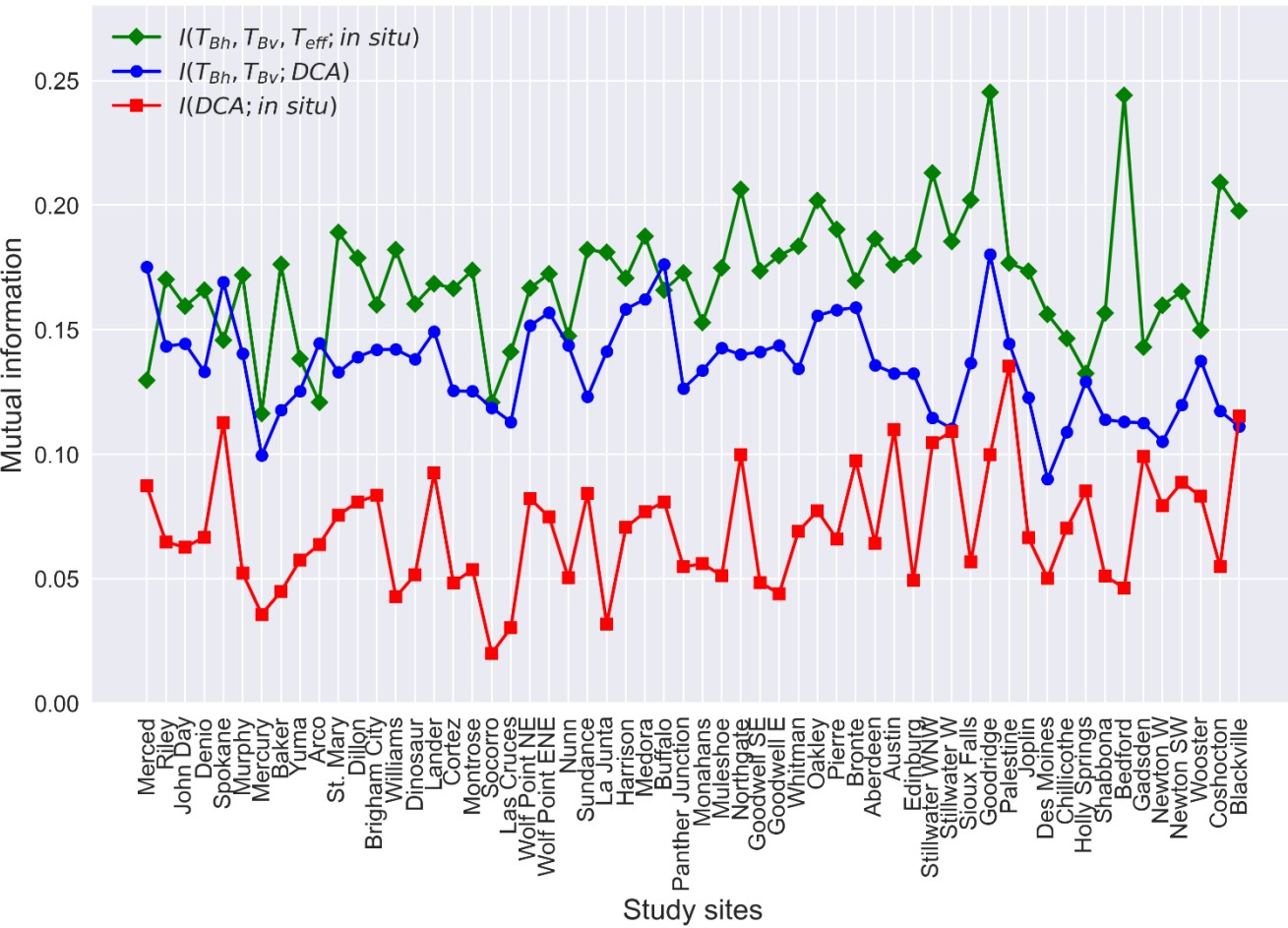

**Figure 3** Mutual information between horizontally polarized brightness temperature ($T_{Bh}$), vertically polarized brightness temperature ($T_{Bv}$), soil effective temperature ($T_{eff}$) and *in situ* soil moisture; mutual information between horizontally polarized brightness temperature ($T_{Bh}$), vertically polarized brightness temperature ($T_{Bv}$) and DCA soil moisture; mutual information between DCA soil moisture and *in situ* soil moisture. See figure 2 caption for site ordering.

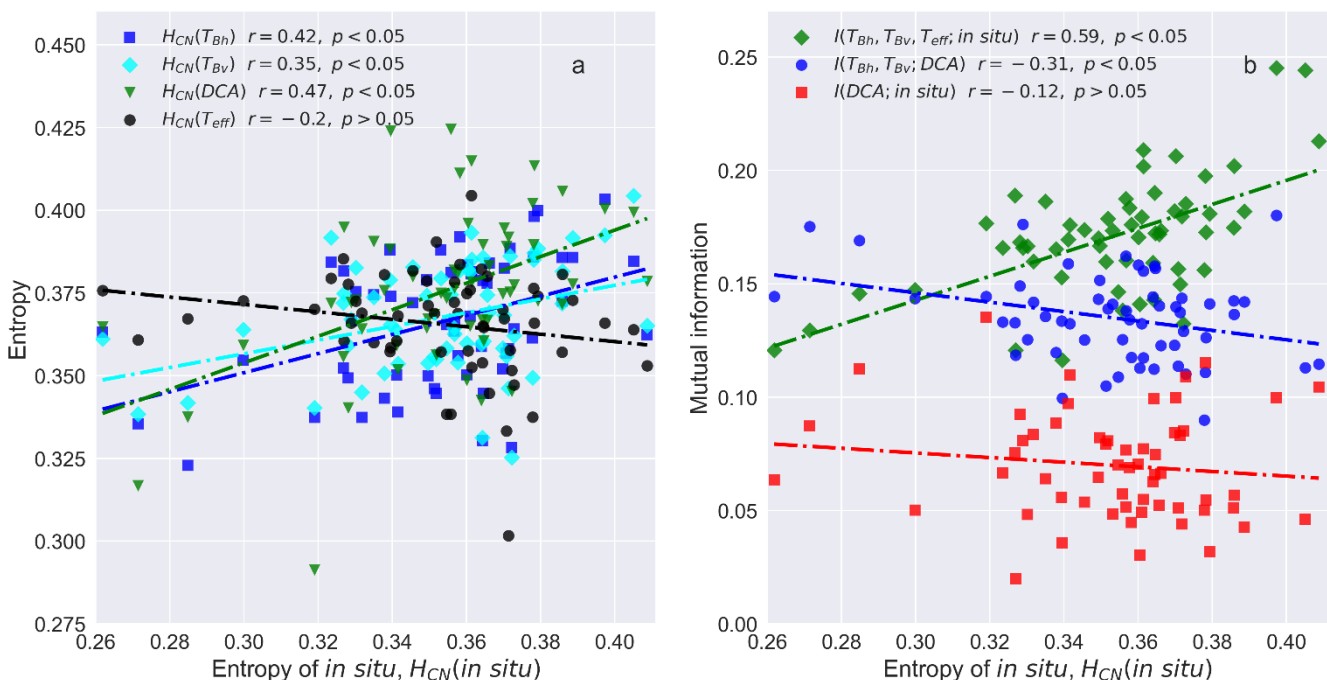

**Figure 4** Entropy of *in situ* soil moisture against the entropies of DCA soil moisture, horizontally polarized brightness temperature ($T_{Bh}$), vertically polarized brightness temperature ($T_{Bv}$) and soil effective temperature ($T_{eff}$) (a) and mutual information quantities (b).

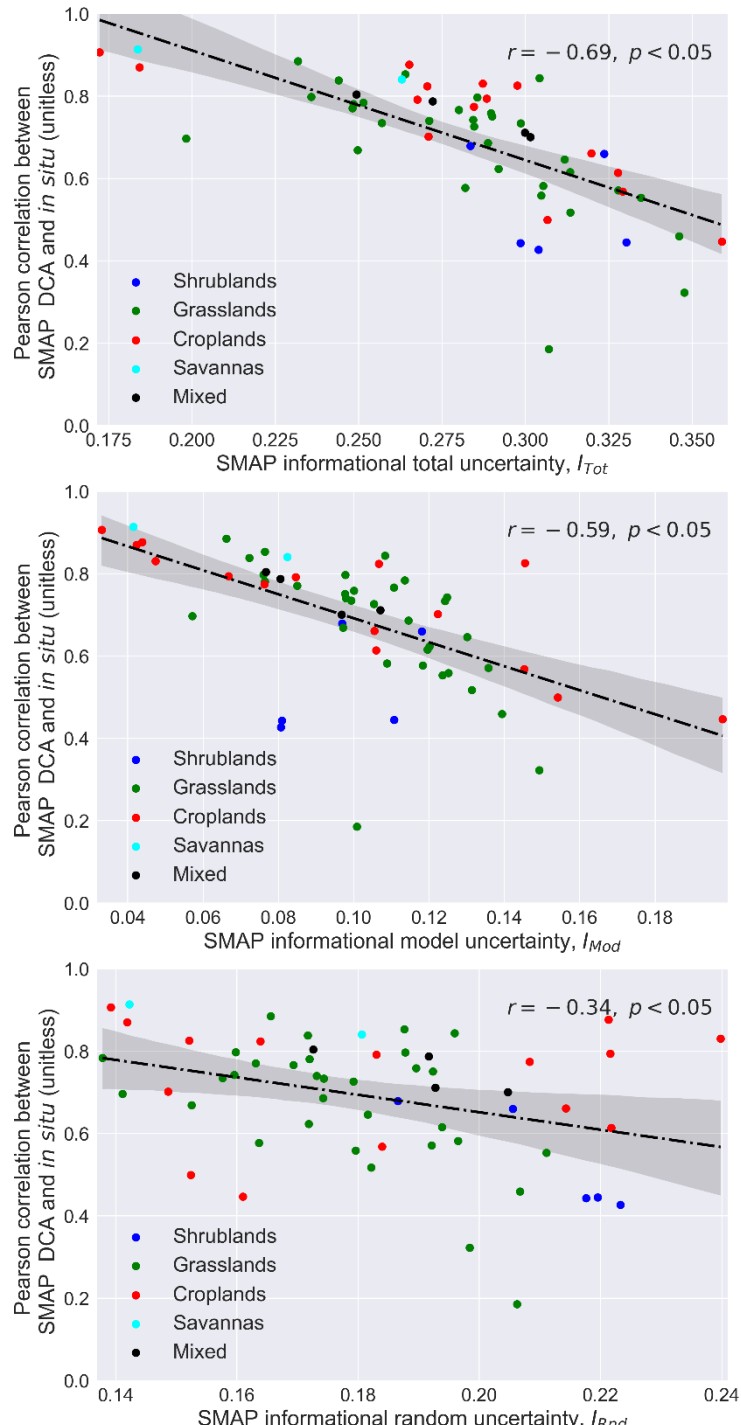


**Figure 5** SMAP informational total uncertainty (a), SMAP informational model uncertainty (b) and SMAP informational random uncertainty against Pearson correlation between *in situ* soil moisture and DCA soil moisture

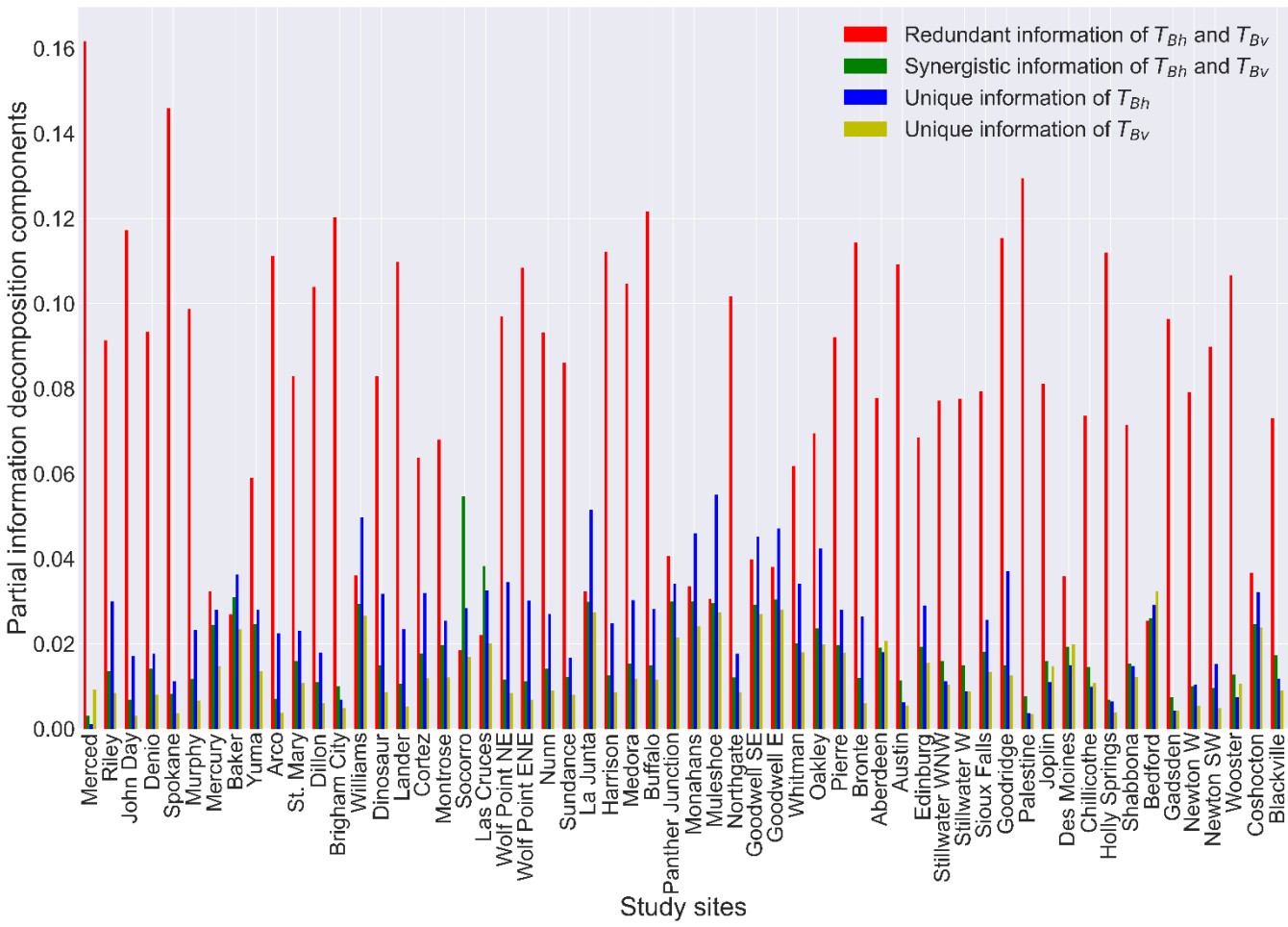

**Figure 6** Partial information decomposition components between horizontally ($T_{Bh}$) and vertically ($T_{Bv}$) polarized brightness temperature and DCA soil moisture. See figure 2 caption for site ordering.




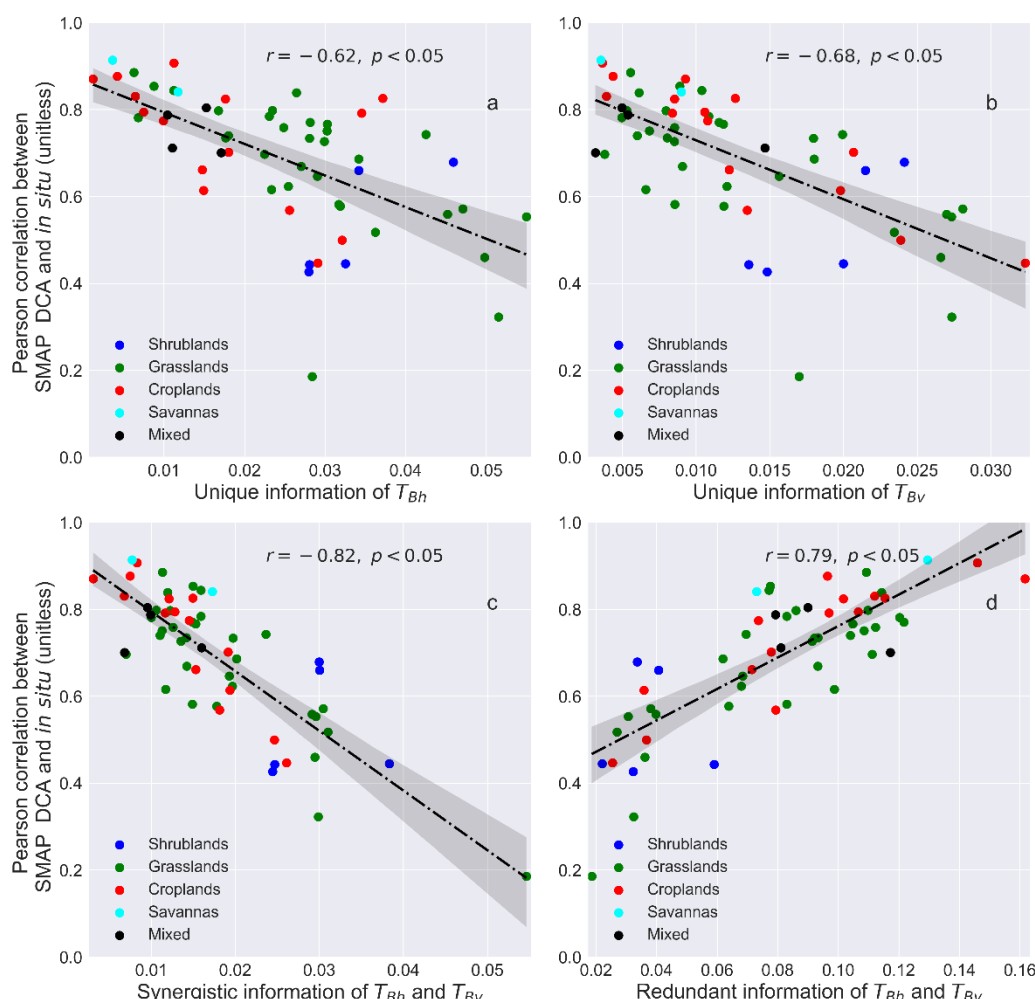

**Figure 7** Partial information decomposition components between horizontally ($T_{Bh}$) and vertically ($T_{Bv}$) polarized brightness temperature against Pearson correlation coefficient between *in situ* and DCA soil moisture.

| Landcover | Informational random uncertainty, $I_{Rnd}$ (and its % of $I_{Tot}$) | Informational model uncertainty, $I_{Mod}$ (and its % of $I_{Tot}$) | Informational total uncertainty, $I_{Tot}$ (and its % of $H_{CN}(in\ situ)$) | Number of Sites |
|---|---|---|---|---|
| Shrublands | 0.21 (68%) | 0.10 (32%) | 0.31 (87%) | 5 |
| Grasslands | 0.18 (63%) | 0.11 (37%) | 0.28 (81%) | 32 |
| Croplands | 0.18 (66%) | 0.10 (34%) | 0.28 (78%) | 15 |
| Savannas | 0.16 (73%) | 0.06 (27%) | 0.22 (64%) | 2 |
| Mixed | 0.19 (68%) | 0.09 (32%) | 0.28 (79%) | 4 |
| Lumped | 0.14 (46%) | 0.17 (54%) | 0.32 (90%) | 58 |
| Overall | 0.18 (65%) | 0.10 (35%) | 0.28 (80%) | 58 |

**Table 1** The amount of informational uncertainties in percentage. The values in the table are the average of each landcover. The values in "Overall" are the average of all the sites. The "Lumped" field is computed using all available dataset.






| Landcover | Unique information of $T_{Bh}$ ($U_h$) (and its % $I(T_{Bh}, T_{Bv}; DCA)$) | Unique information of $T_{Bv}$ ($U_v$) (and its % $I(T_{Bh}, T_{Bv}; DCA)$) | Synergistic information of $T_{Bh}$ and $T_{Bv}$ ($S$) (and its % $I(T_{Bh}, T_{Bv}; DCA)$) | Redundant information of $T_{Bh}$ and $T_{Bv}$ ($R$) (and its % $I(T_{Bh}, T_{Bv}; DCA)$) | Mutual information ($I(T_{Bh}, T_{Bv}; DCA)$) | Number of sites |
|---|---|---|---|---|---|---|
| Shrublands | 0.034 (28%) | 0.019(16%) | 0.029 (25%) | 0.038 (31%) | 0.120 | 5 |
| Grasslands | 0.028 (20%) | 0.013 (10%) | 0.019 (14%) | 0.080 (56%) | 0.140 | 32 |
| Croplands | 0.018 (13%) | 0.013 (10%) | 0.014 (11%) | 0.089 (65%) | 0.134 | 15 |
| Savannas | 0.008 (7%) | 0.006 (5%) | 0.012 (10%) | 0.101 (78%) | 0.128 | 2 |
| Mixed | 0.013(11%) | 0.007 (6%) | 0.011 (9%) | 0.092 (74%) | 0.123 | 4 |
| Lumped | 0.014 (19%) | 0.019 (25%) | 0.008 (11%) | 0.034 (45%) | 0.076 | 58 |
| Overall | 0.024 (18%) | 0.013 (10%) | 0.018 (14%) | 0.080 (58%) | 0.135 | 58 |

**Table 2** The partial information decomposition components. The values in the table are the average of each landcover. The values in "Overall" is the average of all the sites. The "Lumped" field is computed using all available dataset.



