# Peer review of "Information - based uncertainty decomposition in dual channel microwave remote sensing of soil moisture"

_Hydrology and Earth System Sciences, 2020_

## Referee Comment (RC1) · Anonymous Referee #1 · 1 Dec 2020

General comment

This paper presents a performance analysis using information theory to better understand the characterisation potential of the data (TB) and the performance of the inversion algorithms (MCDA). To my knowledge it is a very original approach in the field of application which is targeted and the approach seems to be very relevant. As a naive reader with regard to the analysis method used, I had a little difficulty to follow the details of the calculation (some quantities would gain to be defined), but the essence of the method is well restored and allows a non-specialist reader to understand the approach. My main criticism lies in the scope of the data used to make the analysis.

Indeed, 58 data sets corresponding to 58 stations located in the USA are treated independently. It seems useful to me to recall that the MCDA method aims at exploiting the H and V polarisations in order to separate the reflectivity of the soil from the scattering phenomena linked to vegetation and roughness, the latter being represented by the difference between Tbh and Tbv. By working locally, the variability (humidity, vegetation) is only partially taken into account, taking into account only the annual variations which at the scale of a SMAP pixel present small variations. In fact, by limiting ourselves to a stationary analysis, we underestimate the interest of the MCDA algorithm which is applicable everywhere and allows an estimation of humidity whatever the vegetation cover. This leads to find that the quality of the estimates (here seen by the correlation coefficient between the moisture retrieved and the observed moisture) is all the better as the redundancy term is high, a criterion which is proposed for the following analysis of the quality of the algorithm. The interpretation of R could be better described in the material and method and in particular it is important to specify if a good model is characterized by large value of R, meaning that the model outputs and its input data are well interdependent. The largest R is probably found in low vegetation situations where the ranges of moisture and Tb are greatest. This is a known feature and it seems to me that the quality of the MCDA model is more in its ability to represent the diversity of ecosystems and the associated plant formations. Would it be possible to process a data set of all the stations?

Detailed Comments L85 I think that part of uncertainty is due to the scale of the pixel with mixed surface and in situ moisture that is sparsely sampled (here I think it is local measurement) while the moisture is stongly variable within the pixel.

Eq 5, I suggest to the equation I(TB H or V; Yobs) which is used In equations 7 and 8. It would help me to follow the text

Eq 9 : RMMI is not defined

L209 : an explanation how to interpret The quantity in the context of the study. A good

model should lead to high or low values of U, R and S. At least for S which is the most commented quantity;

L245 : I(h, v ; in situ)? rather than I(MCDA, insitu)

L245 and 247 : honestly i don't see where 0.88 and 0.12% come from. Not evident to see such values in Fig3

L251 : what are the fraction of model uncertainty

L261;263 : how I cand tale 0.55 of I

L264 Uv likely takes greater value if data from different sites are merged

L266:268: yes but at local scale only. Independence of H and V will be much stronger when different location with different ecosystem are taken into account

L279:280 : Interpretation of R should be better explained. I did not see interest of evaluating the quality of retrieval by taking the correlation between H and V into account. It is the negation of the MCDA model that takes the synergy of H and V. This synergy is expressed by taking different sites concurrently, staying at the local scale reducing the interest of having H and V

L303:304 : not only : see comment on L85

L312:315 : What are the parameter considered (tau is derived from H an V) here

L315:317 : speculative ? references

L332:332 : I am not you can say date. The correlation between H and V is well known, the expected ortogonality is more on V-H and H, that is expressed using various ecosystems. Here we are lacking interpretation key. But correlation between inputs does not means that inputs and output are redundant, which my understanding of R

L355 : making the analysis on individual station is a strong limitation, as MSDA capacity were not fully analysed

L358:361 : speculative (reference – difficult to understands without additional information)

L370 : I don't what is the HESS policy. It would be better to have codes in open repository

Figure 2 : remove MI in Y legend

Figure 7 d : the y axis of the embedded graph is not described. The interest of th H V correlation is really limited (see comment above). I suggest to remove it.

In conclusion beside the minor improvement suggested in my comment I expect the authors: 1) better defining the interpretation scheme of the R S and U quantity 2) extending the analyse to merged data set, or at least a subset gathering sites having contrasted ecosystems. This will give stronger overview of the MCDA models and its interes. This might have an impact on the discussion and conclusion.

---

## Referee Comment (RC2) · Anonymous Referee #2 · 7 Dec 2020

The topic of the manuscript is certainly important and interesting for HESS readers. However, the manuscript contains a large number of typos, things that should be explained in more detail and assumptions that are not discussed and could affect the results.

The authors cite (Lines 84-86) just two of the possible uncertainty sources of the retrieval model, however the method discussed in this manuscript is based on in situ measurements. Therefore, the uncertainty of in situ measurements must be taken into account, and it is not at all. In situ measurements are here considered as "ground truth". Unfortunately "ground truth" does not exist, as all measurements, they have

[Figure]

**HESSD**

errors. But probably more important is the uncertainty of the spatial representativeness (satellite Tbs are representative of a spatial scale of tens of kilometer while in situ measurements are single point measurements) and depth representativeness (sensors measure at a given depth while the Tbs are representative of a different/changing depth). These effects must be mentioned and discussed and their possibles effects on the results should be analyzed.

In addition, why using 9km Tbs instead of the original Tbs in the 36 km grid which is closer to the instrument resolution ($\sim$ 50 km). The SM dataset that is provided in a grid with 9-km sampling has been obtained using a Backus-Gilbert interpolation. Surprisingly this is not mentioned at all in the manuscript. How could this choice affect the results as this is another uncertainty source that is not taken into account?

If MDCA is better (at least taking into account that using together Tbh and Tbv adds 15 % of information) why SCA is the official SMAP algorithm and gives better results ? "There is strong interest in the MDCA approach because of its independent estimation of vegetation water status". I probably agree, but this if very very challenging using a single incidence angle. SMOS can do it because it provides multi-incidence angle Tbs. Konings et al. "How Many Parameters Can Be Maximally Estimated From a Set of Measurements?," in IEEE Geoscience and Remote Sensing Letters, vol. 12, no. 5, pp. 1081-1085 have already explained that not because there are two measurements it is possible to actually retrieve two parameters.

Other comments ——————

Line 18: raw data here is undefined. The authors should be more specific so that the abstract is self-explicative

Line 21: "inadequacy" is not a scientific term. What is that inadequacy? Where does it come from ?

Line 67: Peggy O'Neill et al. should be O'Neill et al.

[Figure]

Line 79: 0.04 m3/m3 accuracy target? Which is the metric the authors refer to ?

Line 136: The tau-omega model is not inverted at all. It is used as a forward model and the modeled Tbs are compared to the observed ones varying parameters such as SM. When the Tb's are similar to the observed ones, SM is assumed to be close the real value. There is no inversion of the model giving SM as a function of Tb.

Line 144: it is the uncertainty or the variable that is denoted as H(Yobs) ?

Line 150: The following sentence is meaningless "Although the detailed structure of best achievable model performance maybe remain unknown, mutual information, denoted as I(XInputs; Yobs) where XInputs are the available inputs and Yobs is the in situ measured variable of interest, can provide a good benchmark measure". Please, rephrase.

Line 167: Eq. 2, what is the sense of writing an inequality comparing "mutual informations" (I) with the uncertainty of the variable of interest (H(Yobs))? H and I should not be in the same inequality.

Furthermore, in the example of Eq.1 X is Tbs, Y is Ymodel and Z is Yobs as i) one measures the Tbs, ii) apply the model, iii) Compare to "ground truth". Therefore I(X,Y) >= I(X,Z) should be I(X_inputs, Ymodel) >= I(X_inputs, Yobs) instead of what is written in Eq. 2

Lines 175-180. The manuscript will be clearer if it is stated how to compute those quantities from the actual SM time series records (taking into account the uncertainties)

Line 193 Eq. 5 Why the "mutual information" is compared to uncertainties? Why uncertainties are assumed to be additive ?

Line 195: Eq. 5 expresses I as a function of HCN, how Hcn(Ymdca, Yobs) could be estimated by replacing anything in Eq. 5. Do the authors mean I(Ymdca, Yobs) can be ..." ?

Eq 10. What is -II ?

—- Typos Line 193 Eq. 5 TBv should be T_{B_v}.

Line 177. "is" and "the" are lacking. "Where p IS THE probability..."

Line 194: ... and H_CN() ARE the estimated joint ENTROPIES that ...

Line 196. It IS worth

Eq 8: U_1 should be U_2

Line 232: H_CN(h,v) should be H_CN(Tbh, Tbv)

Line 345. Please correct "theoretic"

---

## Referee Comment (RC3) · Anonymous Referee #3 · 9 Dec 2020

The paper by Li and Good tackles a very important problem of trying to understand the contributions of the sources (observations and model) of uncertainty in SMAP soil moisture retrieval. In general I found the paper easy to read, typographic errors not withstanding, and as a non-expert in information theory I followed the logic of the arguments well. However, as an avid user of SMAP products, I would have like to have seen some attempt to translate the findings into the soil moisture units (mˆ3/mˆ3) and discussion of how the findings may be useful when next I process large time series of the SM estimates.

Specific comments:

[Figure]

Clarify the denominator in Eq. (4)

Scale disparity between in situ and image pixels resolution is not well addressed and I dare say a major contributor to the uncertainty. The conclusion that 88% of the uncertainty is attributable to uncertainty in Tb is a little hard to accept.

L251-258, Fig. 4, and L347-350: This was confusing and can do with greater clarification to aid in the interpretation of the results. As I read it, the fraction of model-to-overall uncertainty is negatively correlated with the cor(in situ,MDCA), while positively correlated with error(in istu,MDCA). What does this mean and what are the implications for model refinement?

---

## Author Comment (AC1) · 21 Jan 2021

Dear anonymous referee #1:

We thank you for the comments that were very insightful to improve our manuscript. We highlighted our replies in blue after each original comment in below. The sentences or paragraphs that were added to the revised manuscript are in red.

General comment

This paper presents a performance analysis using information theory to better understand the characterization potential of the data (TB) and the performance of the inversion algorithms (MCDA). To my knowledge it is a very original approach in the field of application which is targeted and the approach seems to be very relevant. As a naïve reader with regard to the analysis method used, I had a little difficulty to follow the details of the calculation (some quantities would gain to be defined), but the essence of the method is well restored and allows a non-specialist reader to understand the approach. My main criticism lies in the scope of the data used to make the analysis Indeed, 58 data sets corresponding to 58 stations located in the USA are treated independently. It seems useful to me to recall that the MCDA method aims at exploiting the H and V polarizations in order to separate the reflectivity of the soil from the scattering phenomena linked to vegetation and roughness, the latter being represented by the difference between Tbh and Tbv. By working locally, the variability (humidity, vegetation) is only partially taken into account, taking into account only the annual variations which at the scale of a SMAP pixel present small variations. In fact, by limiting ourselves to a stationary analysis, we underestimate the interest of the MCDA algorithm which is applicable everywhere and allows an estimation of humidity whatever the vegetation cover. This leads to find that the quality of the estimates (here seen by the correlation coefficient between the moisture retrieved and the observed moisture) is all the better as the redundancy term is high, a criterion which is proposed for the following analysis of the quality of the algorithm. The interpretation of R could be better described in the material and method and in particular it is important to specify if a good model is characterized by large value of R, meaning that the model outputs and its input data are well interdependent. The largest R is probably found in low vegetation situations where the ranges of moisture and Tb are greatest. This is a known feature and it seems to me that the quality of the MCDA model is more in its ability to represent the diversity of ecosystems and the associated plant formations. Would it be possible to process a data set of all the stations?

Response: We thank the reviewers for these overall constructive comments concerning about this work. Following the reviewer's suggestion, we have partitioned our study sites into different landcovers. The results after partitioned our study sites into different landcovers are shown Table 1 and Table 2 below. We found that Additionally, we switched the 9km SMAP datasets to 36km SMAP datasets to address the comments from another reviewer who would like to know how the choice of different resolution of SMAP products may affect the overall analysis. Therefore, we obtained the 36km SMAP product and we found that the newly obtained 36km SMAP product no longer provides the MDCA soil moisture and is replaced by the Dual Channel Algorithm (DCA) soil moisture with some data updates. Thus, we decided to switch to the newest 9km and 36km SMAP data products. We also included the soil effective temperature ($T_{eff}$) in the uncertainty decomposition analysis because it constitutes a non-trivial information component of the model. We found that there's no pronounced difference between 9km product and 36km product as shown in Figure 1 below ($p > 0.05$, based on two sample t-test). While there is no set in-stone interpretation of the redundant components, we have expanded our description of this aspect of our study. Generally, it should be interpreted with respect to a specific system. For the SMAP DCA, we found that higher $R$ is an indication of better model performances (better Pearson correlation between *in situ* and DCA soil moisture). Finally, more equations were provided in the revised manuscript regarding how we computed each of the quantity involved into this analysis.

[Figure]

Figure 1 Figure 1. Informational uncertainty comparisons between 36km and 9km SMAP DCA products

Detailed Comments L85 I think that part of uncertainty is due to the scale of the pixel with mixed surface and in situ moisture that is sparsely sampled (here I think it is local measurement) while the moisture is strongly variable within the pixel.

Response: We agree and admit that part of the uncertainty is due to scale mismatch between point measurements of *in situ* and SMAP data product. We have added the following "It is important to acknowledge that we used the point based *in situ* soil moisture as the ground truth in this analysis. Due to course spatial resolution of SMAP products, we acknowledge that *in situ* soil moisture may not be able to represent the spatial averaged soil moisture well. Although the nominal sensing depth of L-band SMAP soil moisture is 5 cm, the penetration depth was found to be even shallower in wetter regions (Shellito et al., 2016). In fact, the L-band sensing depth was found to as little as ~1cm (Jackson et al., 2012) and can be more sensitive to surface meteorological conditions and more random than the actual *in situ* soil moisture. The heterogeneity in each pixel relative to the *in situ* observations together with the sensing depth disparity may negatively influence the results of this study result in an overestimate the actual informational uncertainties. We also acknowledge the existence of upscaling methods for matching the *in situ* soil moisture to satellite footprint (Crow et al., 2012). However, most of upscaling methods are achieved under the assistance of additional reference soil moisture datasets. This process introduces additional pieces of information in the DCA system making the separation of the uncertainty induced by the upscaling algorithm or additional dataset from other informational uncertainties much harder. Additionally, we used the hourly *in situ* data to best match the SMAP DCA soil moisture retrievals in time (within an hour). Therefore, it is hard to find such reference dataset at such a high frequency time domain. Here we consider the informational uncertainty caused by the spatial mismatch and sensing depth mismatch between *in situ* and DCA soil moisture as part of the informational random uncertainty ($I_{Rnd}$). Because the DCA essential is a mathematical function and does not inherently requires the inputs of a specific resolution. The spatial resolution is often the inherent attribute of the data. The sensing depth is more of imperfection L-band sensor. The reader should also keep these in mind while interpreting and adopting the results in this study." to the methodology to address this aspect."

Eq 5, I suggest to the equation I(TB H or V; Yobs) which is used In equations 7 and 8. It would help me to follow the text

Response: We thank the reviewer for this comment. We did not change the representation $I(T_{Bh}, T_{Bv}; Y_{obs})$ as suggested in the revised manuscript because we wish to follow standard mathematical notation for this quantities. The reason is that the $I(A, B; C)$ represents the information of random variable A and B together (as a set of random variables {A, B}) shared with the random variable C. The notation in the manuscript follows the notation in other information studies and it also follow the convention (Cover and Thomas, 2005) and other information studies in earth sciences (Goodwell and Kumar, 2017a, 2017b). The notation proposed by the reviewer may interpreted

differently since the "or" means the information specifically from $T_{Bh}$ or $T_{Bv}$ but it this information should be jointly in the notation of the manuscript.

Eq 9: RMMI is not defined
Response: We thank the reviewer for pointing out this. We now have added definition of $R_{MMI}$ as
$$R_{MMI} = \min(I(T_{Bh}; DCA), I(T_{Bv}; DCA))$$

L209: an explanation how to interpret The quantity in the context of the study. A good model should lead to high or low values of U, R and S. At least for S which is the most commented quantity;
Response: We thank the reviewer for this comment. In the context of this study, we found that $R$ is has the largest values and mostly closely related to the performance of SMAP DCA. Therefore, we conclude that a good DCA model/performances should corresponded to higher values of $R$. Therefore, it is expected that higher $R$ should also correspondent to smaller values of $S$, $U_h$ and $U_v$. We have added "Good DCA model performance (as measured by Pearson correlation between *in situ* and DCA soil moisture) is more likely to be found in locations where the redundant information of brightness temperatures shared with DCA soil moisture is high and is more dominant relative to other components." to the revised manuscript.

L245: I(h, v ; in situ)? rather than I(MCDA, insitu)
Response: We thank the reviewer for this comment. In the original manuscript. Line 245 "The information gap between $H_{CN}(in\ situ)$ and $I(MDCA; in\ situ)$ is the overall SMAP uncertainty in which 88% is contributed by the random uncertainty in the systems explanatory variables (Fig. 3)" The abbreviations in the original sentence is correct since the overall SMAP uncertainty is defined as $H_{CN}(in\ situ)$ - $I(MDCA; in\ situ)$. In order to avoid this type confusion, the following equations and paragraphs were added to the manuscript
"

$$I_{Rnd} = H_{CN}(in\ situ) - I(T_{Bh}, T_{Bv}, T_{eff}; in\ situ), \tag{1}$$

$$I_{Mod} = I(T_{Bh}, T_{Bv}, T_{eff}; in\ situ) - I(DCA; in\ situ), \tag{2}$$

and

$$I_{Tot} = H_{CN}(in\ situ) - I(DCA; in\ situ) = I_{Rnd} + I_{Mod}. \tag{3}$$

where $I_{Rnd}$ is the informational random uncertainty, $I_{Mod}$ is the informational model uncertainty, $I_{Tot}$ is the informational total uncertainty, $H_{CN}(in\ situ)$ is the entropy of *in situ* soil moisture, $I(T_{Bh}, T_{Bv}, T_{eff}; in\ situ)$ is the mutual information between horizontally ($T_{Bh}$)- and vertically-polarized brightness temperature ($T_{Bv}$), $I(DCA; in\ situ)$ is the mutual information between DCA soil moisture and *in situ* soil moisture."

L245 and 247: honestly i don't see where 0.88 and 0.12% come from. Not evident to see such values in Fig3
Response: We thank the reviewer for this comment. We have replaced the Figure 3 of the original manuscript with the figure below. A new table that contains these summary statistics is provided (Table 1 below).

[Figure]

Figure 1. Entropy of *in situ* soil moisture against the entropies of DCA soil moisture, horizontally polarized brightness temperature ($T_{Bh}$), vertically polarized brightness temperature ($T_{Bv}$) and soil effective temperature ($T_{eff}$) (a) and mutual information quantities (b)

| Landcover | Informational random uncertainty, $I_{Rnd}$ (and its % of $I_{Tot}$) | Informational model Uncertainty, $I_{Mod}$ (and its % of $I_{Tot}$) | Informational total uncertainty, $I_{Tot}$ (and its % of $H_{CN}(in\ situ)$) |
|---|---|---|---|
| Shrublands | 0.22 (69%) | 0.10 (31%) | 0.32 (88%) |
| Grasslands | 0.20 (62%) | 0.09 (38%) | 0.29 (83%) |
| Croplands | 0.18 (65%) | 0.10 (35%) | 0.28 (79%) |
| Mixed | 0.20 (68%) | 0.09 (32%) | 0.29 (81%) |
| Overall | 0.18 (64%) | 0.11 (36%) | 0.29 (82%) |

**Table 1** The amount of informational uncertainties in percentage. The values in the table are the average of each landcover. The values in "Overall" is the average of all the sites.

L251: what are the fraction of model uncertainty

Response: we thank the reviewer for this comment. In the original manuscript, we mean the proportion of model uncertainty to the overall uncertainty. We have corrected this in the revised manuscript.

L261;263 : how I cand tale 0.55 of I

Response: We thank the reviewer for this comment. We have provided a table for these summary statistics in the revised manuscript (Table 2 below)

| Landcover | Unique information of $T_{Bh}$ ($U_h$) (and its % $I(T_{Bh}, T_{Bv}; DCA)$) | Unique information of $T_{Bv}$ ($U_v$) (and its % $I(T_{Bh}, T_{Bv}; DCA)$) | Synergistic information of $T_{Bh}$ and $T_{Bv}$ ($S$) (and its % $I(T_{Bh}, T_{Bv}; DCA)$) | Redundant information of $T_{Bh}$ and $T_{Bv}$ ($R$) (and its % $I(T_{Bh}, T_{Bv}; DCA)$) | Mutual information ($I(T_{Bh}, T_{Bv}; DCA)$) |
|---|---|---|---|---|---|
| Shrublands | 0.03 (27%) | 0.017(15%) | 0.03 (26%) | 0.036 (32%) | 0.113 |
| Grasslands | 0.029 (21%) | 0.014 (10%) | 0.02 (14%) | 0.077 (55%) | 0.14 |
| Croplands | 0.017 (12%) | 0.013 (9%) | 0.016 (12%) | 0.095 (67%) | 0.141 |
| Mixed | 0.014 (12%) | 0.007 (6%) | 0.01 (8%) | 0.091(74%) | 0.122 |
| Overall | 0.026 (19%) | 0.013 (10%) | 0.019 (14%) | 0.08 (57%) | 0.137 |

**Table 2** The partial information decomposition components. The values in the table are the average of each landcover. The values in "Overall" is the average of all the sites.

L264 Uv likely takes greater value if data from different sites are merged

Response: We thank the reviewer for this comment. The statistics of $U_v$ are shown in Table 2. We found that $U_v$ is consistently the smallest while compared with other components

L266:268: yes but at local scale only. Independence of H and V will be much stronger when different location with different ecosystem are taken into account

Response: We thank the reviewer for this comment. We have extended this analysis to some contrasted landcovers in the revised manuscript.

L303:304 : not only : see comment on L85

Response: We thank the reviewer for this comment. We have added a new paragraph in methodology to address this issue.

L312:315: What are the parameter considered (tau is derived from H an V) here

Response: We thank the reviewer for this comment. We consider the parameter such as vegetation single scattering albedo ($\omega$), surface height standard deviation $s$ etc. We have specified these parameters in the revised manuscript.

L315:317: speculative ? references

Response: We thank the reviewer for this comment. The study from (Konings et al., 2017) has been added as the reference.

L332:332: I am not you can say date. The correlation between H and V is well known, the expected ortogonality is more on V-H and H, that is expressed using various ecosystems. Here we are lacking interpretation key. But correlation between inputs does not means that inputs and output are redundant, which my understanding of R.

Response: We thank the reviewer for this comment. We have removed such statements in the revised manuscript.

L355: making the analysis on individual station is a strong limitation, as MSDA capacity were not fully analysed

Response: We thank the reviewer for the comment. The analysis has been extended to different landcovers in the revised manuscript.

L358:361: speculative (reference – difficult to understands without additional information)

Response: We thank the reviewer for this comment. We found that it may confuse the reader without providing specific information. Therefore, we decided to remove such statements.

L370: I don't what is the HESS policy. It would be better to have codes in open repository

Response: We thank the reviewer for the comment. The python codes and datasets used in this study has been upload to https://github.com/libonancaesar/HESS_Information_Uncertainty.

Figure 2 : remove MI in Y legend

Response: Removed as suggested.

Figure 7d : the y axis of the embedded graph is not described. The interest of th H V correlation is really limited (see comment above). I suggest to remove it.

Response: We thank the reviewer for this comment. The embedded graph has been removed as suggested.

In conclusion beside the minor improvement suggested in my comment I expect the authors: 1) better defining the interpretation scheme of the R S and U quantity 2) extending the analyse to merged data set, or at least a subset gathering sites having contrasted ecosystems. This will give stronger overview of the MCDA models and its interes. This might have an impact on the discussion and conclusion.

Response: We thank the reviewer for this comment. (1) the definition of these components has been defined in the methodology of the revised manuscript (2) Additional analysis regarding different landcovers has been added

References

Cover, T. M. and Thomas, J. A.: Elements of Information Theory, Wiley., 2005.

Crow, W. T., Berg, A. A., Cosh, M. H., Loew, A., Mohanty, B. P., Panciera, R., de Rosnay, P., Ryu, D. and Walker, J. P.: Upscaling sparse ground-based soil moisture observations for the validation of coarse-resolution satellite soil moisture products, Rev. Geophys., 50(2), 634, doi:10.1029/2011RG000372, 2012.

Goodwell, A. E. and Kumar, P.: Temporal information partitioning: Characterizing synergy, uniqueness, and redundancy in interacting environmental variables, Water Resour. Res., 53(7), 5920–5942, doi:10.1002/2016WR020216, 2017a.

Goodwell, A. E. and Kumar, P.: Temporal information partitioning: Characterizing synergy, uniqueness, and redundancy in interacting environmental variables, Water Resour. Res., doi:10.1002/2016WR020216, 2017b.

Jackson, T. J., Bindlish, R., Cosh, M. H., Zhao, T., Starks, P. J., Bosch, D. D., Seyfried, M., Moran, M. S., Goodrich, D. C., Kerr, Y. H. and Leroux, D.: Validation of Soil Moisture and Ocean Salinity (SMOS) Soil Moisture Over Watershed Networks in the U.S., IEEE Trans. Geosci. Remote Sens., 50(5), 1530–1543, doi:10.1109/TGRS.2011.2168533, 2012.

Konings, A. G., Piles, M., Das, N. and Entekhabi, D.: L-band vegetation optical depth and effective scattering albedo estimation from SMAP, Remote Sens. Environ., 198, 460–470, doi:10.1016/j.rse.2017.06.037, 2017.

Shellito, P. J., Small, E. E., Colliander, A., Bindlish, R., Cosh, M. H., Berg, A. A., Bosch, D. D., Caldwell, T. G., Goodrich, D. C., McNairn, H., Prueger, J. H., Starks, P. J., van der Velde, R. and Walker, J. P.: SMAP soil moisture drying more rapid than observed in situ following rainfall events, Geophys. Res. Lett., 43(15), 8068–8075, doi:10.1002/2016GL069946, 2016.

---

## Author Comment (AC2) · 21 Jan 2021

Dear anonymous referee #2:

Thank you for providing such valuable suggestions and comments on our manuscript. Please find the response to these comments (in blue). The sentences or paragraphs that were added to the revised manuscript are in red.

The topic of the manuscript is certainly important and interesting for HESS readers. However, the manuscript contains a large number of typos, things that should be explained in more detail and assumptions that are not discussed and could affect the results. The authors cite (Lines 84-86) just two of the possible uncertainty sources of the retrieval model, however the method discussed in this manuscript is based on in situ measurements. Therefore, the uncertainty of in situ measurements must be taken into account, and it is not at all. In situ measurements are here considered as "ground truth". Unfortunately "ground truth" does not exist, as all measurements, they have errors. But probably more important is the uncertainty of the spatial representativeness (satellite Tbs are representative of a spatial scale of tens of kilometer while in situ measurements are single point measurements) and depth representativeness (sensors measure at a given depth while the Tbs are representative of a different/changing depth). These effects must be mentioned and discussed and their possible effects on the results should be analyzed.

Response: We thank the reviewers for these overall constructive comments. We agree and admit that part of the uncertainty is due to scale mismatch between point measurements of *in situ* and SMAP data product. We also acknowledged that the sensing depth of the SMAP may vary, though the designated sensing depth is up to 5cm. In practice, the sensing depth may be even shallower. We have classified the uncertainty induced by the sensing depth and spatial mismatch as part of the informational random uncertainty. This is because the model does not contain the resolution itself and the uncertainty induced by sensing depth is more of imperfection of the sensor. We added the following paragraph "It is important to acknowledge that we used the point based *in situ* soil moisture as the ground truth in this analysis. Due to course spatial resolution of SMAP products, we acknowledge that *in situ* soil moisture may not be able to represent the spatial averaged soil moisture well. Although the nominal sensing depth of L-band SMAP soil moisture is 5 cm, the penetration depth was found to be even shallower in wetter regions (Shellito et al., 2016). In fact, the L-band sensing depth was found to as little as ~1cm (Jackson et al., 2012) and can be more sensitive to surface meteorological conditions and more random than the actual *in situ* soil moisture. The heterogeneity in each pixel relative to the *in situ* observations together with the sensing depth disparity may negatively influence the results of this study and result in an overestimate the actual informational uncertainties. We also acknowledge the existence of upscaling methods for matching the *in situ* soil moisture to satellite footprint (Crow et al., 2012). However, most of upscaling methods are achieved under the assistance of additional reference soil moisture datasets. This process introduces additional pieces of information in the DCA system making the separation of the uncertainty induced by the upscaling algorithm or additional dataset from other informational uncertainties much harder. Additionally, we used the hourly *in situ* data to best match the SMAP DCA soil moisture retrievals in time (within an hour). Therefore, it is hard to find a reference dataset at with high frequency in time domain and good spatial coverage. Here we consider the informational uncertainty caused by the spatial mismatch and sensing depth mismatch between *in situ* and DCA soil moisture as part of the informational random uncertainty ($I_{Rnd}$). Because the DCA essential is a mathematical function and does not inherently require the inputs to be at a specific resolution. The spatial resolution is often the inherent attribute of the data. The sensing depth is more of imperfection L-band sensor. The reader should also keep these in mind while interpreting and adopting the results in this study." to the revised manuscript

In addition, why using 9km Tbs instead of the original Tbs in the 36 km grid which is closer to the instrument resolution (_ 50 km). The SM dataset that is provided in a grid with 9-km sampling has been obtained using a Backus-Gilbert interpolation. Surprisingly this is not mentioned at all in the manuscript. How could this choice affect the results as this is another uncertainty source that is not taken into account?

Response: We thank the reviewer for this comment. We compared informational uncertainty from both the 9km SMAP datasets and 36km SMAP datasets to address the resolution effects. We found that the newly obtained 36km SMAP product no longer provides the MDCA soil moisture and is replaced by the Dual Channel Algorithm (DCA) soil moisture with some data updates when we were obtaining the 36km SMAP product. Thus, we decided to switch to the newest 9km and 36km SMAP data product for the comparisons of resolution effects. We have also included the soil effective temperature ($T_{eff}$) in the uncertainty decomposition analysis since this information is important in the DCA modeling process. We found that the differences in informational uncertainties between SMAP DCA 36km and 9km product are not pronounced (Figure 1 below). The results from a two-sample t-test between SMAP 9km and SMAP 36km information uncertainties shown that there is no significant different between SMAP 9km and SMAP 36km in informational uncertainties ($p > 0.05$). Given no pronounced resolution effects on informational uncertainties, we decided to proceed by using the original SMAP 36km product for this study.

[Figure]

Figure 1. Informational uncertainty comparisons between 36km and 9km SMAP DCA products

If MDCA is better (at least taking into account that using together Tbh and Tbv adds 15 % of information) why SCA is the official SMAP algorithm and gives better results ? "There is strong interest in the MDCA approach because of its independent estimation of vegetation water status". I probably agree, but this if very very challenging using a single incidence angle. SMOS can do it because it provides multi-incidence angle Tbs. Konings et al. "How Many Parameters Can Be Maximally Estimated From a Set of Measurements?," in IEEE Geoscience and Remote Sensing Letters, vol. 12, no. 5, pp. 1081-1085 have already explained that not because there are two measurements it is possible to actually retrieve two parameters.

Response: We thank the reviewer for this comment. We did not state that the MDCA/DCA perform better than the SCA. The objective of our study is to partition the overall informational uncertainty into the uncertainty caused by the DCA input data streams and that caused by the model itself. We also thank the reviewer for providing this valuable paper reference. We mentioned the dual channel is interesting not only because it provides soil moisture but provided vegetation optical depth estimation that cannot be independently estimated through the SCAs. It is reasonable to assume that the vegetation optical depth may not be accurate as there are large uncertainties in the DCA. Hence, finding where the information is lost (informational uncertainties) in the DCA can be helpful for DCA soil moisture estimations and hence more accurate estimation of vegetation optical depth. We thank the reviewer for providing this valuable reference and have cited this reference in the following "There is strong interest in the DCA approach because of its independent estimation of vegetation opacity in lieu of the specified vegetation climatology employed by the SCA. Additionally, it has been suggested that using a time-integrated vegetation opacity, as is employed in the multi-temporal dual channel algorithm (MT-DCA) for instance (Piles et al., 2016), improves the estimates of soil and vegetation state. These contrasting approaches, as well as other studies on SMAP's temporal polarized ratio algorithm (TPRA) (Gao et al., 2020) and regularized dual channel algorithm (RDCA) (Chaubell et al., 2019), suggested there is still uncertainty about how SMAP observations of horizontal

and vertical brightness temperature can be best translated into estimates of surface properties. Although SMAP can provide spatially explicit soil moisture estimates that have been shown to be useful to understand a set of ecohydrological problems (Jadidoleslam et al., 2019), the soil moisture retrievals are still subject to significant amount of uncertainty due to the imperfection of the model and the forcing datasets. The success of retrieving soil moisture and vegetation opacity are interdependent (Konings et al., 2017) and it is important to consider the how the amount of duplicate information carried within a set of observations limits the number of parameters to be inferred (Konings et al., 2015). Therefore, it is critical to diagnosis and quantify the causality of the uncertainty caused by the SMAP algorithm in order to improve the soil moisture and vegetation opacity retrieval quality." of the introduction

Other comments —————

Line 18: raw data here is undefined. The authors should be more specific so that the abstract is self-explicative

Response: We thank the reviewer for this comment. The "raw data" has been explicitly replaced by "$T_{Bh}$, $T_{Bv}$ and $T_{eff}$" that are the inputs to the DCA.

Line 21: "inadequacy" is not a scientific term. What is that inadequacy? Where does it come from?

Response: We thank the reviewer for this comment. We have replaced "inadequacy" with the term "a lack of additional explanatory power beyond $T_{Bh}$, $T_{Bv}$ and $T_{eff}$" in the revised manuscript.

Line 67: Peggy O'Neill et al. should be O'Neill et al.

Response: We thank the reviewer for the comment. The citation style has been corrected as suggested.

Line 79: 0.04 m3/m3 accuracy target? Which is the metric the authors refer to ?

Response: We thank the reviewer for this comment. The metric that we are referring to is ubRMSE. We have specified the metric in the revised manuscript.

Line 136: The tau-omega model is not inverted at all. It is used as a forward model and the modeled Tbs are compared to the observed ones varying parameters such as SM. When the Tb's are similar to the observed ones, SM is assumed to be close the real value. There is no inversion of the model giving SM as a function of Tb.

Response: We thank the reviewer for this comment. This sentence has been rephrased as "It requires the brightness temperatures as the main inputs, soil effective temperature as an ancillary input, and is parameterized based on overlaying vegetation and soil surface information. The DCA iteratively feeds the 'tau-omega' model with initial guesses of soil moisture and vegetation optical depth." in the revised manuscript.

Line 144: it is the uncertainty or the variable that is denoted as H(Yobs) ?

Response: We thank the reviewer for this comment. We have dropped this term in the revised manuscript and have provide a more explicit definition.

Line 150: The following sentence is meaningless "Although the detailed structure of best achievable model performance maybe remain unknown, mutual information, denoted as I(XInputs; Yobs) where XInputs are the available inputs and Yobs is the in situ measured variable of interest, can provide a good benchmark measure". Please, rephrase.

Response: We thank the reviewer for this comment. The above statement has been rephrased to "Mutual information between the model inputs and *in situ* observations of model output can be used as a useful and effective measure of best achievable performance model because it links the model inputs and *in situ* observations only

through the joint and marginal probability mass functions that do not involve any priori model assumptions (Gong et al., 2013)."

Line 167: Eq. 2, what is the sense of writing an inequality comparing "mutual informations" (I) with the uncertainty of the variable of interest (H(Yobs))? H and I should not be in the same inequality.

Response: We thank the reviewer for this comment. We found that these inequalities may introduce unnecessary confusions to the readers. Therefore, we have dropped this equation (1) and equation (2) in the revised manuscript. The explanation is that the entropy $H(.)$ can be interpreted as the uncertainty inherent in a random variable or the amount of information requires to describe a random variable. The maximum information of another or other explanatory random variables can provided/capture the information about such random variable should be the entropy of this random variable.

Furthermore, in the example of Eq.1 X is Tbs, Y is Ymodel and Z is Yobs as i) one measures the Tbs, ii) apply the model, iii) Compare to "ground truth". Therefore I(X,Y) >= I(X,Z) should be I(X_inputs, Ymodel) >= I(X_inputs, Yobs) instead of what is written
in Eq. 2

Response: We thank the reviewer for this comment. We have dropped this equation. The explanation of the inequalities can be found in this paper (Gong et al., 2013).

Lines 175-180. The manuscript will be clearer if it is stated how to compute those quantities from the actual SM time series records (taking into account the uncertainties)

Response: We thank the reviewer for this comment. The details about how we calculated entropy, mutual information and informational uncertainties in the DCA systems has been provided in the revised manuscript.

Line 193 Eq. 5 Why the "mutual information" is compared to uncertainties? Why
uncertainties are assumed to be additive?

Response: We thank the reviewer for this comment. The reason why uncertainties is assumed to be additive is because the way informational random uncertainty and informational model uncertainty are defined in (Gong et al., 2013). We have provided the following descriptions "For a given system in which the inputs and output are linked via mathematical functions, the mutual information between model outputs and *in situ* observation can never exceed the entropy of the *in situ* observations. This information gap is defined as informational total uncertainty ($I_{Tot}$). The mutual information between the *in situ* observations and the available explanatory variables is also always smaller than the entropy of *in situ* observations. This information gap, defined as informational random uncertainty ($I_{Rnd}$), is caused by the existence of inherent data uncertainty of the explanatory variables and a lack of complete explanatory variables to fully capture the information in the *in situ* observations." in the revised manuscript.

Line 195: Eq. 5 expresses I as a function of HCN, how Hcn(Ymdca, Yobs) could be estimated by replacing anything in Eq. 5. Do the authors mean I(Ymdca, Yobs) can be ..." ?

Response: We thank the reviewer for this comment. We have added the following equations to the revised manuscript "

$$H(\mathrm{X}, \mathrm{Y}) = -\sum_{x \in X} \sum_{y \in Y} p(x, y) log_2 \, p(x, y), \qquad (1)$$

where $p(x, y)$ is the joint probability mass function associated with X and Y that is estimated by the same method mentioned above. The same normalization and correction method of eq. (2) is applied to joint entropy of eq. (3). The entropy after the correction and normalization is formulated as

$$H_{CN}(\mathrm{X}, \mathrm{Y}) = \frac{H(\mathrm{X},\mathrm{Y}) + \frac{K-1}{2n}}{log_2 \, n}, \qquad (2)$$

where $H_{CN}(X, Y)$ is the corrected and normalized joint entropy of random variable associated with $\{X, Y\}$, $H(X, Y)$ is the uncorrected entropy from eq. (3), $n$ is the number of data points that were used to calculate the normalized joint entropy (hereafter joint entropy), $K$ is the number of non-zero joint probabilities based on the Freeman and Diaconis method (Freedman and Diaconis, 1981). All the joint entropies that are associated with two or more random variables in the later equations (i.e., $H_{CN}(in\ situ, DCA)$, $H_{CN}(T_{Bh}, T_{Bv}, DCA)$, $H_{CN}(T_{Bh}, T_{Bv}, T_{eff}, in\ situ)$ etc.) are computed using the combination of eq. (3) and eq. (4) with the replacement of $p(\bullet)$ by their joint probability mass functions, respectively. "

Eq 10. What is -II ?

Response: We thank the reviewer for this comment. II is the interaction information and -II is the negative number of II.

—- Typos Line 193 Eq. 5 TBv should be T_{B_v}.

Response: We thank the reviewer for this comment. We have corrected the typo.

Line 177. "is" and "the" are lacking. "Where p IS THE probability..."

Response: We thank the reviewer for this comment. We have corrected the typo.

Line 194: ... and H_CN() ARE the estimated joint ENTROPIES that ...

Response: We thank the reviewer for this comment. We have corrected the typo.

Line 196. It IS worth

Response: We thank the reviewer for this comment. We have corrected the typo.

Eq 8: U_1 should be U_2

Response: We thank the reviewer for this comment. We have corrected the typo.

Line 232: H_CN(h,v) should be H_CN(Tbh, Tbv)

Response: We thank the reviewer for this comment. We have corrected this typo.

Line 345. Please correct "theoretic"

Response: We thank the reviewer for this comment. We have corrected this.

References:

Chaubell, J., Yueh, S., Chan, S., Dunbar, S., Colliander, A., Entekhabi, D. and Chen, F.: Smap Regularized Dual-Channel Algorithm for the Retrieval of Soil Moisture and Vegetation Optical Depth, in IGARSS 2019 - 2019 IEEE International Geoscience and Remote Sensing Symposium, pp. 5312–5315, IEEE., 2019.

Crow, W. T., Berg, A. A., Cosh, M. H., Loew, A., Mohanty, B. P., Panciera, R., de Rosnay, P., Ryu, D. and Walker, J. P.: Upscaling sparse ground-based soil moisture observations for the validation of coarse-resolution satellite soil moisture products, Rev. Geophys., 50(2), 634, doi:10.1029/2011RG000372, 2012.

Gao, L., Sadeghi, M., Ebtehaj, A. and Wigneron, J.-P.: A temporal polarization ratio algorithm for calibration-free retrieval of soil moisture at L-band, Remote Sens. Environ., 249, 112019, doi:10.1016/j.rse.2020.112019, 2020.

Gong, W., Gupta, H. V., Yang, D., Sricharan, K. and Hero, A. O.: Estimating epistemic and aleatory uncertainties during hydrologic modeling: An information theoretic approach, Water Resour. Res., 49(4), 2253–2273,

doi:10.1002/wrcr.20161, 2013.

Jackson, T. J., Bindlish, R., Cosh, M. H., Zhao, T., Starks, P. J., Bosch, D. D., Seyfried, M., Moran, M. S., Goodrich, D. C., Kerr, Y. H. and Leroux, D.: Validation of Soil Moisture and Ocean Salinity (SMOS) Soil Moisture Over Watershed Networks in the U.S., IEEE Trans. Geosci. Remote Sens., 50(5), 1530–1543, doi:10.1109/TGRS.2011.2168533, 2012.

Jadidoleslam, N., Mantilla, R., Krajewski, W. F. and Goska, R.: Investigating the role of antecedent SMAP satellite soil moisture, radar rainfall and MODIS vegetation on runoff production in an agricultural region, J. Hydrol., 579, 124210, doi:10.1016/j.jhydrol.2019.124210, 2019.

Konings, A. G., McColl, K. A., Piles, M. and Entekhabi, D.: How Many Parameters Can Be Maximally Estimated From a Set of Measurements?, IEEE Geosci. Remote Sens. Lett., 12(5), 1081–1085, doi:10.1109/LGRS.2014.2381641, 2015.

Konings, A. G., Piles, M., Das, N. and Entekhabi, D.: L-band vegetation optical depth and effective scattering albedo estimation from SMAP, Remote Sens. Environ., 198, 460–470, doi:10.1016/j.rse.2017.06.037, 2017.

Piles, M., Entekhabi, D., Konings, A. G., McColl, K. A., Das, N. N. and Jagdhuber, T.: Multi-temporal microwave retrievals of Soil Moisture and vegetation parameters from SMAP, in 2016 IEEE International Geoscience and Remote Sensing Symposium (IGARSS), pp. 242–245, IEEE., 2016.

Shellito, P. J., Small, E. E., Colliander, A., Bindlish, R., Cosh, M. H., Berg, A. A., Bosch, D. D., Caldwell, T. G., Goodrich, D. C., McNairn, H., Prueger, J. H., Starks, P. J., van der Velde, R. and Walker, J. P.: SMAP soil moisture drying more rapid than observed in situ following rainfall events, Geophys. Res. Lett., 43(15), 8068–8075, doi:10.1002/2016GL069946, 2016.

---

## Author Comment (AC3) · 21 Jan 2021

Dear anonymous referee #3:

Thank you for your valuable comments. Below we respond (highlighted in blue) to the reviewer's comments. The text that were added to the revised manuscript are marked in red.

The paper by Li and Good tackles a very important problem of trying to understand the contributions of the sources (observations and model) of uncertainty in SMAP soil moisture retrieval. In general I found the paper easy to read, typographic errors not withstanding, and as a non-expert in information theory I followed the logic of the arguments well. However, as an avid user of SMAP products, I would have like to have seen some attempt to translate the findings into the soil moisture units (m^3/m^3) and discussion of how the findings may be useful when next I process large time series of the SM estimates.

Response: We thank the reviewers for these overall constructive comments concerning about this work. Unfortunately, this study cannot translate the information quantities into the specific soil moisture unit of ($m^3/m^3$). We did not observe a strong relationship between the RMSE and the information uncertainties/decomposed mutual information components, likely in part to the large differences in the soil moisture regimes at different sites. RMSE as an absolute metric is more sensitive to these absolute differences while the correlation coefficient, being normalized by the variance at the site itself is less sensitive and more comparable across sites. The redundant components can be very helpful for large time series processing. For instance, the larger the redundant components of $T_{Bh}$, $T_{Bv}$ to the DCA algorithm, the more likely we can obtain high quality datasets. The tolerance of the data quality depends on the user's need. In general, redundant information between $T_{Bh}$, $T_{Bv}$ to the DCA of a value greater than 0.1 can be indicative of a better overall retrieval quality (~0.75 in Pearson correlation). We have added the following to the revised manuscript "Given the strength of this relationship, the $R$ could be potentially used as a DCA evaluation metric that doesn't depend on *in situ* measurement and ancillary dataset. It is also useful for SMAP DCA soil moisture users to have a rough estimation of how high the quality of the obtained DCA soil moisture without actually knowing the *in situ* soil moisture. However, this depends on specific user requirements for data quality. In general, the DCA soil moisture tends to be in high end in term retrieval quality (~ 0.75 in Pearson correlation) when the $R$ is greater 0.1.".

Specific comments:
Clarify the denominator in Eq. (4)
Response: We thank the reviewer for this comment. We have clarified this in the revised manuscript.

Scale disparity between in situ and image pixels resolution is not well addressed and I dare say a major contributor to the uncertainty. The conclusion that 88% of the uncertainty is attributable to uncertainty in Tb is a little hard to accept.
Response: We thank the reviewer for this comment and now better clarify where this estimate comes from. We have switched the 9km SMAP datasets to 36km SMAP datasets to address the comments from another reviewer who would like to know how the choice of different resolution of SMAP products may affect the overall analysis. Therefore, we obtained the 36km SMAP product and we found that the newly obtained 36km SMAP product no longer provides the MDCA soil moisture and is replaced by the Dual Channel Algorithm (DCA) soil moisture with some data updates. Thus, we decided to switch to the newest 9km and 36km SMAP data product. We have also found that we did not included the soil effective temperature ($T_{eff}$) in the uncertainty decomposition analysis. Hence, the results from the updated manuscript are now based on the consideration of soil effective temperature. In general, we found that 64% percent of the information total uncertainty is caused by informational random uncertainty from the input datasets of DCA. For now, we have classified the uncertainty induced by the sensing depth and spatial mismatch as part of the informational random uncertainty. This is because the model does not contain the resolution itself and the uncertainty induced by sensing depth is more of imperfection of the sensor. However, it is extremely

hard to sperate what's the proportion of informational random uncertainty is specifically caused by spatial mismatch. We have added the following text in the revised manuscript "It is important to acknowledge that we used the point based *in situ* soil moisture as the ground truth in this analysis. Due to course spatial resolution of SMAP products, we acknowledge that *in situ* soil moisture may not be able to represent the spatial averaged soil moisture well. Although the nominal sensing depth of L-band SMAP soil moisture is 5 cm, the penetration depth was found to be even shallower in wetter regions (Shellito et al., 2016). In fact, the L-band sensing depth was found to as little as ~1cm (Jackson et al., 2012) and can be more sensitive to surface meteorological conditions and more random than the actual *in situ* soil moisture. The heterogeneity in each pixel relative to the *in situ* observations together with the sensing depth disparity may negatively influence the results of this study and result in an overestimate the actual informational uncertainties. We also acknowledge the existence of upscaling methods for matching the *in situ* soil moisture to satellite footprint (Crow et al., 2012). However, most of upscaling methods are achieved under the assistance of additional reference soil moisture datasets. This process introduces additional pieces of information in the DCA system making the separation of the uncertainty induced by the upscaling algorithm or additional dataset from other informational uncertainties much harder. Additionally, we used the hourly *in situ* data to best match the SMAP DCA soil moisture retrievals in time (within an hour). Therefore, it is hard to find a reference dataset at with high frequency in time domain and good spatial coverage. Here we consider the informational uncertainty caused by the spatial mismatch and sensing depth mismatch between *in situ* and DCA soil moisture as part of the informational random uncertainty ($I_{Rnd}$). Because the DCA essential is a mathematical function and does not inherently require the inputs to be at a specific resolution. The spatial resolution is often the inherent attribute of the data. The sensing depth is more of imperfection L-band sensor. The reader should also keep these in mind while interpreting and adopting the results in this study."

L251-258, Fig. 4, and L347-350: This was confusing and can do with greater clarification to aid in the interpretation of the results. As I read it, the fraction of model-to-overall uncertainty is negatively correlated with the cor(in situ,MDCA), while positively correlated with error(in istu,MDCA). What does this mean and what are the implications for model refinement?

Response: We thank the reviewer for this comment. As we mentioned earlier, we changed the SMAP product in order to address the comment from the other reviewer. In the analysis of this new product, we found that there is no significant correlation between the RMSE and correlation between *in situ* soil moisture and DCA soil moisture. Therefore, we decided to exclude the RMSE plots. We have also plotted the actual informational model uncertainty against the Pearson correlation of *in situ* and DCA soil moisture. The implications of this analysis for model refinements are (1): more robust water body correction methods are needed for SMAP brightness temperature observations. The quality of the model effective soil temperature that goes to the SMAP DCA system need to be further evaluated (2): model uncertainties can be reduced potentially by a better parameterization scheme such as replacing time independent parameter with seasonal dependent parameters especially in locations where there are seasonal changes in landcover or vegetation phenology

References
Crow, W. T., Berg, A. A., Cosh, M. H., Loew, A., Mohanty, B. P., Panciera, R., de Rosnay, P., Ryu, D. and Walker, J. P.: Upscaling sparse ground-based soil moisture observations for the validation of coarse-resolution satellite soil moisture products, Rev. Geophys., 50(2), 634, doi:10.1029/2011RG000372, 2012.
Jackson, T. J., Bindlish, R., Cosh, M. H., Zhao, T., Starks, P. J., Bosch, D. D., Seyfried, M., Moran, M. S., Goodrich, D. C., Kerr, Y. H. and Leroux, D.: Validation of Soil Moisture and Ocean Salinity (SMOS) Soil Moisture Over Watershed Networks in the U.S., IEEE Trans. Geosci. Remote Sens., 50(5), 1530–1543, doi:10.1109/TGRS.2011.2168533, 2012.

Shellito, P. J., Small, E. E., Colliander, A., Bindlish, R., Cosh, M. H., Berg, A. A., Bosch, D. D., Caldwell, T. G., Goodrich, D. C., McNairn, H., Prueger, J. H., Starks, P. J., van der Velde, R. and Walker, J. P.: SMAP soil moisture drying more rapid than observed in situ following rainfall events, Geophys. Res. Lett., 43(15), 8068–8075, doi:10.1002/2016GL069946, 2016.

---

## Author Response (AR2)

Dear anonymous referee #1:

We thank you for providing such thoughtful comments to improve our manuscript. The responses of the comments are highlighted in blue. The new paragraphs/sentences that were added to the revised manuscript are marked in red. As an additional note to the reviewer, after re-evaluation of our datasets, we now have more sites included in this revised manuscript (58 sites now compared to 51 in the last version). This is because analysis of the last version was focused on the sites where both SMAP 9km dataset and SMAP 36km datasets pass the quality control and we only used the 36km dataset. But here in this version, we focus on 36km data product. Therefore, more sites are available as some sites did not have 9km data. In addition, after re-checking the time of DCA observations, we found that the DCA brightness temperature time field were slightly miss aligned in our prior analysis by ~12hrs. We have corrected such mistake in this version. These adjustments have not signifyingly altered our results or conclusions drawn in this paper.

Compared to the previous version, the methodology is presented in a clearer way.

1.  The fact that the results are treated by type of surface makes it possible to enrich the discussion. However, these groupings do not respond to the suggestion made, i.e. to treat all the surfaces together for the calculation of entropy in order to properly reflect the model's capacity to treat different surfaces. By calculating the parameters site by site and comparing the results to local statistics one can arrive at a wrong interpretation of the statistics. I find it counter-intuitive that the best results are obtained when the information is redundant rather than synergistic. One explanation for this is that the best correlation is obtained on surfaces where the vegetation cover is less important (with larger range of value and little vegetation effects). This is also the situation where Tbh and TBv are the most correlated, hence the high R. To say that R can be a good indicator to evaluate a model is a bit hazardous in this case. A comment on this point should be made in the discussion.

Response: We have now also computed the uncertainties by lumping all the datasets together and have shown the results in "Lumped" column of table 1 in the revised manuscript. In addition, we also did the partial information decomposition using the "Lumped" dataset. The results are shown in revised version of table 2. We explored the relationship between these information components and different metrics such as vegetation density, vegetation homogeneity, informational total uncertainty etc. We found that these quantities are all marginally related to the informational components ($S$, $R$, $U_h$, $U_v$). We think high $R$ (lower $S$) is both physically affected by vegetation and how the algorithm processes the $T_{Bh}$ and $T_{Bv}$. Therefore, our $S$ and $R$ is an integration of these factors. We have added "These information components were found to be marginally correlated with factors such as vegetation density (the Pearson correlation of average LAI with $R, S, U_h U_v$ are 0.23, -0.38, -0.54, and -0.19 respectively) and vegetation heterogeneity (the Pearson correlation of LAI standard deviation with $R, S, U_h, U_v$ are 0.22, -0.39, -0.54, and -0.22 respectively). Additionally, these informational components were also found to be correlated with the mutual information shared between brightness temperatures and DCA estimates (the Pearson correlation of $I(T_{Bh}, T_{Bv}; DCA)$ with $R, S, U_h, U_v$ are 0.6, -0.28, 0.22, and -0.16 respectively), the informational total uncertainty (the Pearson correlation of $I_{Tot}$ with $R, S, U_h, U_v$ are -0.76, 0.62, 0.56, and 0.68 respectively), informational random uncertainty (the Pearson correlation of $I_{Rnd}$ with $R, S, U_h, U_v$ are -0.42 , 0.29, 0.05, and 0.15 respectively), and informational model uncertainty (the Pearson correlation of $I_{Mod}$ with $R, S, U_h, U_v$ are -0.63, 0.56, 0.66, and 0.75 respectively).

This indicates that these informational components in the DCA system are not only physically driven by both vegetation density and heterogeneity but also other factors such as how algorithm processes the information from $T_{Bh}$ and $T_{Bv}$ to produce the DCA outputs. It is more likely to observe higher $R$ and lower $S$ in locations where vegetation is denser and more heterogeneous, yet the correlation of these variables with model quality (0.47 for mean LAI and 0.42 for the standard deviation of LAI) are weaker than the correlations found between $R$ and $S$ and model quality shown in Figure 7." to 4.2 Model evaluation from another perspective. The following paragraph "In addition, we find the proportion of informational uncertainty increases as the data is lumped together relative to averaging these statics calculated on a site-by-site basis (Table 1). Treating all the surfaces together as a whole does not reduce the informational total uncertainty because the lumping process contains both "high quality" and "low quality" (as assessed by the Pearson correlation between *in situ* and DCA soil moisture) datasets. The uncertainties in these datasets may accumulate while lumping them together and result in an increase in total informational uncertainty." was also added to 4.1 DCA informational uncertainties."

Detailed comments

Comment est calculé teff (a mettre dans la section 2 plutôt que 4)

2.   L241 coarse

Response: We thank the reviewer for this comment. The typo has been corrected.

3.   L244 …more sensitive …. In situ soil moisture: I don't undersdand

Response: We thank the reviewer for the comment. I acknowledge that such statement may confuse the reader. Therefore, we have replaced "can be more sensitive to surface meteorological conditions and more random than the actual *in situ* soil moisture." by "was found to vary wet or dry surface soil conditions (Escorihuela et al., 2010; Raju et al., 1995)".

4.   L244 you can consider the following papers dedicated to sampling depth.

M.-J. Escorihuela, A. Chanzy, J.-P. Wigneron, et Y. H. Kerr, « On the effective soil moisture depth of L-band radiometry: a case study », Remote Sensing of Environment, vol. 114, p. 995 1001, 2010, doi: doi:10.1016/j.rse.2009.12.011.

S. Raju, A. Chanzy, J. P. Wigneron, J. C. Calvet, Y. Kerr, et L. Laguerre, « Soil moisture and temperature profile effects on microwave emission at low frequencies », Remote Sensing of Environment (NLD), vol. 54, no 2, p. 85 97, 1995.

Response: We thank the reviewer for providing these papers. We have cited them in revised paper. Please refer to the response in Comment 3

5.   L246 is it really an overestimate.

Response: We thank the viewer for point out this. We acknowledge this can be overestimate or underestimate. Therefore, we have changed "result in an overestimate the actual informational uncertainties" to "bias the estimation of informational uncertainty" in the revised manuscript.

6.   L294 what you mean by high quality data set?

Response: We thank the reviewer for this comment. The high-quality dataset here is referred to as the high Pearson correlation coefficient between the SMAP DCA soil moisture and *in situ* soil moisture. We have added the following to clarify this "(higher correlation between *in situ* soil moisture

and SMAP DCA soil moisture)".

7.   L305: This indicates ...: see my general comment. The DCA is designed to take profit of TBh and the difference between TBh and TBv. Physically this statement is wrong and probably requires deeper discussion.

Response: We thank the reviewer for this insightful comment. We have removed such statement from the revised manuscript.

8.   L320'321 How can you say that?

Response: We thank the reviewer for such comment. We have replaced such statement "Mutual information can provide a way of unambiguously define the best achievement performance of a model that is able to completely transform the available information to the desired target given a set of the input data" with "It offers an opportunity of partitioning the total informational uncertainty in the DCA to the uncertainty due to the input datasets and the uncertainty due to model structure and model parameterizations. This partition cannot be achieved by leveraging the common DCA assessment metrics (Chan et al., 2018) (e.g., Pearson correlation, ubRMSE) that only involve the DCA soil moisture and *in situ* soil moisture".

9.   L333 why schrubland are less sensitive to water availability.

Response: We thank the reviewer for this comment. We acknowledge that such statement might be erroneous. Therefore, we have removed this sentence.

10.   L374 Tbh and TBv are correlated but Tbv-Tbh give an orthogonal information linked to the roughness and the vegetation. This is the essence of DCA. Taking care of sweeping the physic like this on the basis of your statistic that might be not well posed.

Response: We thank the reviewer for this comment. We have removed this sentence from the paragraph mentioned.

11. For the originality of the approach, the quality of presentation this paper deserve publication but apparent conflict between the physic and the results need a bit more discussion and some final statement might a bit tempered.

Response: We thank the reviewer for supporting our article. We have added the explanation regarding this partial information decomposed components in the discussion and have tempered the overall conclusion in the revised manuscript.

References

Chan, S. K., Bindlish, R., O'Neill, P., Jackson, T., Njoku, E., Dunbar, S., Chaubell, J., Piepmeier, J., Yueh, S., Entekhabi, D., Colliander, A., Chen, F., Cosh, M. H., Caldwell, T., Walker, J., Berg, A., McNairn, H., Thibeault, M., Martínez-Fernández, J., Uldall, F., Seyfried, M., Bosch, D., Starks, P., Holifield Collins, C., Prueger, J., van der Velde, R., Asanuma, J., Palecki, M., Small, E. E., Zreda, M., Calvet, J., Crow, W. T. and Kerr, Y.: Development and assessment of the SMAP enhanced passive soil moisture product, Remote Sens. Environ., 204, 931–941, doi:10.1016/j.rse.2017.08.025, 2018.

Escorihuela, M. J., Chanzy, A., Wigneron, J. P. and Kerr, Y. H.: Effective soil moisture sampling depth

of L-band radiometry: A case study, Remote Sens. Environ., 114(5), 995–1001, doi:10.1016/j.rse.2009.12.011, 2010.

Raju, S., Chanzy, A., Wigneron, J.-P., Calvet, J.-C., Kerr, Y. and Laguerre, L.: Soil moisture and temperature profile effects on microwave emission at low frequencies, Remote Sens. Environ., 54(2), 85–97, doi:10.1016/0034-4257(95)00133-L, 1995.

Dear anonymous referee #2:

We thank you for your valuable comments. The replies to the comments are highlighted in blue. The new text added to the revised manuscript are marked in red. As an additional note to the reviewer, after re-evaluation of our datasets, we now have more sites included in this revised manuscript (58 sites now compared to 51 in the last version). This is because analysis of the last version was focused on the sites where both SMAP 9km dataset and SMAP 36km dataset pass the quality control and we only used the 36km dataset. But here in this version, we focus on 36km data product. Therefore, more sites are available as some sites did not have 9km data. In addition, after re-checking the time of DCA observations, we found that the DCA brightness temperature time field were slightly miss aligned in our prior analysis by ~12hrs. We have corrected such mistake in this version. These adjustments have not significantly altered our results or conclusions drawn in this paper.

The manuscript has improved significantly in readability, the methodology explanation is much clearer in the new version. The analysis per land cover classes is an interesting addition. Using the SMAP data set which is provided in a 36 km, closer to the instrumental resolution, is a good choice as well. A plot was provided in the authors answer to my previous comments comparing the results using the "36 km" or the "9 km" SMAP data sets, showing no significant differences. This is expected as providing a 50 km resolution data set in a 9 km grid or in a 36 km grid should not affect the results of this study.

I have a number of comments:

1.  "Higher redundant information provided by Tbh and Tbv tends to be found in land covers with less woody components". This is surprising, the effet of those woody components is to create a depolarisation making TbH and Tbv more similar. In this sense, I would expect that they are more redundant when the vegetation cover is denser. The manuscript will still improve if such affirmations are interpreted in relation with our physical understanding of the signal. My feeling is that the analysis in terms of "redundancy" is only showing that the for the easiest sites (those with more homogeneous land covers, topography, roughness, meteorological conditions, those for which remote sensing estimations agree the best with in situ measurement), the results are good using either TbH or Tbv. This does not imply that "redundancy" is the reason of the better results. Regarding the affirmation "The informational redundancy between these remotely sensed observations can be used as independent metric to assess the retrieval quality of the algorithms". I do not agree until the physical insight mentioned above is included.

Response: We thank this reviewer for such insightful comment. We found that the redundant component is related to many factors. We explored the relationship between the redundant component ($R$) and site vegetation density and surface vegetation homogeneity as indicated by LAI values the site LAI standard deviation. As suggested by the reviewer, we found that vegetation density is marginally correlated with $R$ (Pearson correlation [$r(\text{LAI}; R)$] of 0.22). In addition, the vegetation homogeneity is also marginally correlation with R ($r(\text{LAI std}; R)$] of 0.23). The $R$ is also correlated with mutual information between $T_{Bh}$, $T_{Bv}$ and SMAP DCA ( $r(I(\text{DCA}; T_{Bh}, T_{Bv}); R)$ of 0.6) as well as the informational model and random uncertainties. Therefore, the metric $R$ is not only integrating information about how surface vegetation (vegetation density and vegetation

homogeneity) may affect the algorithm performance, but also provided insights into how the SMAP DCA processes these the brightness temperature data streams. The correlation between $R$ and the DCA model quality is higher than the correlation of the mean (or std) of LAI with the model quality. Furthermore, the correlation between R and the DCA model quality is also higher than that of the direct correlation of $T_{Bh}$ and $T_{Bv}$ with the model quality. This suggests that the $R$ is more informative, and integrates across, these multiple effects (both physical and computational).

We have added the following paragraph to 4.2 Model evaluation from another perspective. "These information components were found to be marginally correlated with factors such as vegetation density (the Pearson correlation of average LAI with $R, S, U_h U_v$ are 0.23, -0.38, -0.54, and -0.19 respectively) and vegetation heterogeneity (the Pearson correlation of LAI standard deviation with $R, S, U_h, U_v$ are 0.22, -0.39, -0.54, and -0.22 respectively). Additionally, these informational components were also found to be correlated with the mutual information shared between brightness temperatures and DCA estimates (the Pearson correlation of $I(T_{Bh}, T_{Bv}; DCA)$ with $R, S, U_h, U_v$ are 0.6, -0.28, 0.22, and -0.16 respectively), the informational total uncertainty (the Pearson correlation of $I_{Tot}$ with $R, S, U_h, U_v$ are -0.76, 0.62, 0.56, and 0.68 respectively), informational random uncertainty (the Pearson correlation of $I_{Rnd}$ with $R, S, U_h, U_v$ are -0.42 , 0.29, 0.05, and 0.15 respectively), and informational model uncertainty (the Pearson correlation of $I_{Mod}$ with $R, S, U_h, U_v$ are -0.63, 0.56, 0.66, and 0.75 respectively). This indicates that these informational components in the DCA system are not only physically driven by both vegetation density and heterogeneity but also other factors such as how algorithm processes the information from $T_{Bh}$ and $T_{Bv}$ to produce the DCA outputs. It is more likely to observe higher $R$ and lower $S$ in locations where vegetation is denser and more heterogeneous, yet the correlation of these variables with model quality (0.47 for mean LAI and 0.42 for the standard deviation of LAI) are weaker than the correlations found between $R$ and $S$ and model quality shown in Figure 7. The $R$ and $S$ metric in this study can thus not only integrate information about how the surface vegetation density and heterogeneity influence the algorithm performance but provided insight into how effectively DCA algorithm uses the information from $T_{Bh}$ and $T_{Bv}$.

Compared with other ancillary and *in situ* independent metrics such as correlation strength between Pearson correlation of $T_{Bh}$ with $T_{Bv}$ and the Pearson correlation between *in situ* and DCA soil moisture (0.67), the correlation strength of $S$ and $R$ with Pearson correlation of *in situ* and DCA soil moisture are tighter (0.79 and -0.82 for $R$ and $S$). This suggests the complex non-linear relationship between of $T_{Bh}$, $T_{Bv}$ with DCA soil moisture is better captured by $R$ and $S$ as compared to the direct correlation between the two brightness temperatures themselves. Given the strength of this relationship, the $R$ and $S$ holds the potential to be used as a DCA evaluation metric that does not depend on *in situ* measurement and ancillary dataset. It is also useful for SMAP DCA soil moisture users to have a rough estimation of how high the quality (as characterized as the correlation strength between DCA and *in situ*) of the obtained DCA soil moisture without actually knowing the *in situ* soil moisture."

2.  Same for Figure 7. The higher correlations of DCA SM and in situ measurements would be found when the unique information of TBv is the lowest, when the unique information of TBh is the lowest and when the synergistic information is the lowest... could the physical mechanisms be explained?
Response: We thank the reviewer for this comment. We found that these informational components

are correlation with both physical factors such as LAI, the standard deviation of LAI and informational uncertainties. Such relationships are driven by the collected effect of land surface characteristics and how the algorithm process these data streams. Therefore, no single factor can totally explain why the higher correlation of DCA soil moisture and *in situ* soil moisture is more likely to be found in low $U_h$ and $U_v$. Please also refer to the response of comment 2.

3. Regarding physical insight, Fig 4b could be interesting, but it is not commented in the text. The mutual information of TbH, TbV and Teff with respect to in situ do show a correlation with the entropy of the in situ measurements, in contrast to I(DCA, in situ). Could this be interpreted in terms of dynamics of the time series ? More dynamics, more entropy, more information content of TbH, TbV and Teff with respect to in situ ?

We thank the reviewer for this comment. That is a good point. We think the interpretation from the reviewer maybe right. We have added the following to the discussion "As shown in figure 4b, the $I(T_{Bh}, T_{Bv}, T_{eff}; \textit{in situ})$ increase as there are more dynamics in the *in situ* soil moisture which is also reflected by high values of $H_{CN}(T_{Bh})$ and $H_{CN}(T_{Bv})$. The raw observations ($T_{Bv}, T_{Bh},$ and $T_{eff}$) provide more available information to the system, whereas such information is not properly captured by the algorithm as reflected by low correlation strength between $H_{CN}(\textit{in situ})$ and $I(DCA; \textit{in situ})$. Therefore, it is more likely to observe large information model uncertainties where the soil moisture is more dynamic which may cause a low efficiency of DCA to correctly transmit the available information."

4. What Konings et al (2017) actually shows is that not because there are two measurements one can retrieve simultaneously two variables with good accuracy (SM and VOD). SMOS can retrieve both SM and VOD because there are tens of Tb measurements for different incidence angles. This could have been mentioned explicitly in the introduction, and this could justify using DCA instead of MDCA. However, I think that in addition to Tbh, Tbv and Teff, the DCA algorithm use NDVI as input, in contrast to MDCA. What would be the implications for this work of not taking NDVI into account?

Response: We thank the reviewer for such comment. We have removed "The success of retrieving soil moisture and vegetation opacity is interdependent (Konings et al., 2017)". We've also added the following " Other L-band focused satellite mission such as Soil Moisture and Ocean Salinity (SMOS) retrieves both soil moisture and vegetation optical depth by using numerous brightness measurements for different incidence angles (Kerr et al., 2012)." to the introduction.

Vegetation water content climatology is derived from MODIS NDVI and these values, while different for each day of year and location combination, these values do not vary across different years. It is used to estimate the initial guess for the unknown vegetation optical depth. Here we consider this more of a dynamic parameter and not as an input data stream. However this parameter does potentially introduce additional information. We think including NDVI vegetation water content may decrease the estimated informational random uncertainty and increases the informational model uncertainty. We acknowledge this may be one of the limitations of this study. We have added the sentences below to the 4.3 Approach Limitations section "It is important to understand that SMAP DCA system retrieves soil moisture with the help of vegetation water content climatology derived from a MODIS NDVI data stream. This is specified as a set value for each location and day of year combination and is used to estimate the initial guess for the unknown vegetation

optical depth (O'Neill et. al., 2020). The reader should keep in mind that this study considers such data as a dynamic time-varying parameter and it is not treated as a data input in this study. Adding NDVI as a data input would result in $I(T_{Bh}, T_{Bv}, T_{eff}, NDVI; in\ situ)$ being larger than or equal to $I(T_{Bh}, T_{Bv}, T_{eff}; in\ situ)$ in the calculation of $I_{Rnd}$, and therefore $I_{Rnd}$ would decrease. Since, $I_{Tot}$ only considers DCA output and *in situ* data it is not altered by adding dynamic parameters and $I_{Mod}$ would therefore increase. Thus, consideration of additional dynamic parameters in this informational assessment would serve to shift uncertainties from those attributed to the input data themselves to uncertainties attributed to the model structure and parameterization."

5. Abstract Line 23-25: "Quality": please say explicitly which is the quality metric used. A few lines afterwards "Pearson correlations" are mentioned but again nothing is said of the variables used to compute that correlation.

Response: We thank the reviewer for this comment. We have added "denoted as the Pearson correlation coefficient between SMAP DCA soil moisture and *in situ* soil moisture" to the abstract.

6. Line 28: "... than FOR other land covers"

Response: We thank the reviewer for such comment. We no longer have that statement in the revised manuscript.

7. Line 31: "redundancy"

Response: Thank the reviewer for this comment. The wording is corrected as suggested.

8. Line 135: It would be good to add the list of the stations used to ensure the reproductivility of the results

Response: We thank the reviewer for this comment. We have provided a new table (table S1) that contains a list of stations used in this study in the supplementary materials.

9. Line 146: please give explicitly the quality flags and/or thresholds used to filter the data

Response: We thank the reviewer for this comment. We have provided the thresholds and quality flags used for SMAP DCA soil moisture, soil effective temperature and horizontally- and vertically polarized brightness temperature. The following sentences were added "The extracted data series were filtered by the internal quality flags of $T_{Bh}$ ("tb_qual_flag_h"), $T_{Bv}$ ("tb_qual_flag_v") and DCA ("retrieval_qual_flag_option3") as provided in SMAP data files. We retain data points at a particular SMAP observation time when they all simultaneous pass quality control and fall within their correspondent valid ranges (e.g., $0 \sim 330K$ for $T_{Bh}$ and $T_{Bv}$, $253.15K \sim 313.15K$ for $T_{eff}$, $> 0.02m^3/m^3$ for DCA soil moisture) as specified in the SMAP documentation (Chan, 2020)."

10. Line 147: "data points" here refer to time samples or in situ sites ?

Response: We thank the reviewer for this comment. Here we mean the time sample. We have added "We retain data points at a particular SMAP observation time" to the paragraph mentioned. Please also see the reply of comment 8.

11. Lines 270-276: Could you add an intuitive explanation on why the mutual information between model outputs and in situ observations cab never exceed the entropy of in situ observations ?

In terms of Eq 7 this would mean that H_CN(DCA)-H_CN(DCA, insitu) is necessarily negative. But personally, from the information given in the manuscript I still do not understand why.

Response: We thank the reviewer for this comment and have added an intuitive explanation. This reads as follows: "Conceptually, the entropies of model output and *in situ* observations can be considered as two circles (of equal or unequal sizes) and the mutual information between model output and *in situ* observation can be viewed as the overlapping area of these two circles (Uda, 2020). Therefore, the maximum mutual information shared between model output and *in situ* is the minimum of the entropy of model output and *in situ* observations, i.e: $I(DCA, in\ situ) \leq \min[H(DCA), H(in\ situ)]$. Intuitively, the overlapping area of two circles cannot be larger that of the smaller circle. Because we are focused on representing the observed soil condition, the information gap between *in situ* observations, $H(in\ situ)$, and the mutual information shared between *in situ* observations and model output, $I(DCA, in\ situ)$, is defined as informational total uncertainty ($I_{Tot}$). "

12.   Line 354: "The sensing depth is more of imperfection L band sensor". I am afraid I do not agree. The representativeness of the two measurements (in situ or remote sensing) in depth is conceptually of the same characteristics of the spatial representativeness of both measurements. Both in depth or in space, in situ sensors or remote sensor simply measure different things.

Response: We thank the reviewer for this comment. We have removed such sentence from the revised manuscript.

13. Figure 7 x label of panel c: correct "inforamtion"

Response: We thank the reviewer for pointing out this typo. We have corrected this in the revised figure 7c.

References

Chan, S.: Soil Moisture Active Passive (SMAP) Level 2 Passive Soil Moisture Product Specification Document, Jet Propuls. Lab. Inst. Technol. Pasadena, USA, JPL D-72547 (Version 7.0), 63, 2020.

Kerr, Y. H., Waldteufel, P., Richaume, P., Wigneron, J. P., Ferrazzoli, P., Mahmoodi, A., Al Bitar, A., Cabot, F., Gruhier, C., Juglea, S. E., Leroux, D., Mialon, A. and Delwart, S.: The SMOS Soil Moisture Retrieval Algorithm, IEEE Trans. Geosci. Remote Sens., 50(5), 1384–1403, doi:10.1109/TGRS.2012.2184548, 2012.

O'Neill, P., Bindlish, R., Chan, S., Njoku, E., and Jackson, T.: Algorithm theoretical basis document: Level 2 & 3 soil moisture (passive) data products, Lev. 2 3 soil moisture data Prod. Jet Propuls. Lab. Inst. Technol. Pasadena, USA, JPL D-66480 (revision F), 100, 2020.

Uda, S.: Application of information theory in systems biology, Biophys. Rev., 12(2), 377–384, doi:10.1007/s12551-020-00665-w, 2020.

---

## Author Response (AR3)

Dear anonymous referee #2:

We thank you for your insightful comments. The replies to the comments are highlighted in blue. The new text added to the revised manuscript are marked in red.

I still do not find the formulation in Lines 34-35 clear. Redundancy in between TBh and TBv is certainly not the cause of better retrieval quality. But is possible that for surfaces and conditions that make possible good retrievals, TBh and TBv are more redundant than in other cases. In other words, I suggest rephrasing to avoid giving the impression that there is a cause-effect relationship.

Response: We thank the reviewer for this comment. We have rephrased Line 34-Line 35 to "$T_{Bh}$ and $T_{Bv}$ tend to contribute large redundant information to DCA estimates under surfaces or 30 conditions where DCA makes better retrievals."

Otherwise, I only have a minor comment: Line 529. Please note that in the case of SMAP, NDVI is not used as a first guess to estimate the vegetation optical depth (VOD). NDVI is used to fix the VOD value and to only retrieve soil moisture. So, I strongly suggest to remote "initial guess".

Response: We thank the reviewer for this comment. We have removed the "initial guess" from Line 529.